# Endoplasmic reticulum membranes are continuously required to maintain mitotic spindle size and forces

Margarida Araújo[1], Alexandra Tavares[1], Diana V Vieira[1] (iD), Ivo A Telley[1] (iD), Raquel A Oliveira[1,2] (iD)

Membrane organelle function, localization, and proper partitioning upon cell division depend on interactions with the cytoskeleton. Whether membrane organelles also impact the function of cytoskeletal elements remains less clear. Here, we show that acute disruption of the ER around spindle poles affects mitotic spindle size and function in *Drosophila* syncytial embryos. Acute ER disruption was achieved through the inhibition of ER membrane fusion by the dominant-negative cytoplasmic domain of atlastin. We reveal that when centrosome-proximal ER membranes are disrupted, specifically at metaphase, mitotic spindles become smaller, despite no significant changes in microtubule dynamics. These smaller spindles are still able to mediate sister chromatid separation, yet with decreased velocity. Furthermore, by inducing mitotic exit, we found that nuclear separation and distribution are affected by ER disruption. Our results suggest that ER integrity around spindle poles is crucial for the maintenance of mitotic spindle shape and pulling forces. In addition, ER integrity also ensures nuclear spacing during syncytial divisions.

## Introduction

Cell division is often simplified to an isolated process of chromosome segregation by the mitotic spindle (Vitre et al, 2014), followed by scission of the cell membrane during cytokinesis (Mierzwa & Gerlich, 2014). Thereby, the spindle apparatus assembles, attaches chromosomes, aligns them, and generates the force required to pull sister chromatids apart (Petry, 2016; Maiato et al, 2017). However, in compartmentalized eukaryotic cells various organelles undergo extensive reorganization and distribution to the daughter cells during mitosis (Champion et al, 2017; Carlton et al, 2020). One of those organelles is the ER, a tubular network that forms a continuum with the nuclear envelope (Goyal & Blackstone, 2013). When the nuclear envelope breaks down in mitosis, allowing spindle microtubules (MTs) to interact with the chromosomes, the ER reorganizes in the vicinity of the spindle (Bobinnec et al, 2003). ER membranes have recently been proposed to hinder efficient chromosome segregation. When ER membrane biogenesis is increased by enhanced fatty acid synthesis, this leads to higher viscosity of the cytoplasm and ultimately to chromosome missegregation (Merta et al, 2021). Moreover, chromosomes that become unsheathed by the ER are more prone to segregation errors (Ferrandiz et al, 2022). Hence, ER reorganization may be a passive event, required for faithful mitosis and even distribution of this organelle to daughter cells (Smyth et al, 2015). Reciprocally, it has also been proposed that ER reorganization plays a functional role during mitosis. On one hand, the ER enclosing the spindle could act as a molecular exclusion barrier, sorting or concentrating cell cycle and spindle-relevant proteins (Schweizer et al, 2015). On the other hand, it is conceivable that the ER plays a mechanical role and balances spindle forces, thus adjusting, for example, the spindle length (Dumont & Mitchison, 2009). This is mostly supported by the observation that functional disruption, at mitotic entry, of another membranous structure—the nuclear envelope—perturbs spindle assembly (Tsai et al, 2006; Liu & Zheng, 2009; Ma et al, 2009; Civelekoglu-Scholey et al, 2010). At the molecular level, Receptor expression-enhancing proteins (REEPs) were shown to be involved in the exclusion of ER membranes from the spindle region during mitosis (Schlaitz et al, 2013). Furthermore, the ER targeting kinase TAOK2 is important for tethering of ER membranes to the MT cytoskeleton and for ER mobility along MTs during mitosis (Nourbakhsh et al, 2021). Altogether, these studies exposed the role of membranes during mitosis and demonstrated that the association between the ER and MTs is important for spindle assembly. Whether this association is required continuously, even after unperturbed spindle assembly, is currently unknown.

Deciphering an actively contributing versus passively hindering role of the ER during mitosis is critical. However, investigating these potential roles is technically challenging because of the difficulty to perturb such a critical structure for cell physiology in a fast and temporally controlled manner. Here, we used microinjection approaches in *Drosophila* syncytial embryos to acutely disrupt ER membranes in a metaphase-arrested state. We examined mitotic spindle morphology and function upon loss of ER integrity. We uncovered that ER membranes surrounding the spindle pole are

[1]Instituto Gulbenkian de Ciência, Oeiras, Portugal    [2]Universidade Católica Portuguesa, Católica Medical School, Católica Biomedical Research Centre, Lisbon, Portugal

Correspondence: itelley@igc.gulbenkian.pt; rcoliveira@igc.gulbenkian.pt

important for the maintenance of mitotic spindle architecture and forces.

# Results and Discussion

To visualize the ER during *Drosophila* syncytial embryonic divisions, we performed live imaging and followed ER and nuclear envelope localization throughout the cell cycle. We used flies expressing the chromatin marker H2B–mRFP1 and an ER marker with an EYFP flanked by an ER targeting signaling peptide (human calreticulin target sequence) at N-terminus and a KDEL sequence, a small peptide sequence that retains proteins in the ER lumen (Bräuer et al, 2019), at C-terminus, referred therein simply as ERsp-EYFP–KDEL (LaJeunesse et al, 2004). We also used the ER membrane-shaping protein reticulon-like protein 1 (Rtnl1) fused to Green Fluorescence Protein (GFP) (Rtnl1–GFP). We marked MTs by injecting Alexa647-labelled tubulin (Fig 1 and Video 1). ERsp-EYFP-KDEL reports all ER (LaJeunesse et al, 2004), whereas Rtnl1 specifically reports locations of membrane reshaping. Consistent with this notion, we observed an extended EYFP-labelled ER network in the entire cortex of the syncytial embryo and throughout mitosis (Fig 1A), with partial overlap with the nuclear envelope (lamin–GFP, see Fig S1, t = 00:00). This suggests an interaction between these two membranous structures during interphase that changes upon nuclear envelope breakdown. Note that in contrast to the canonical "open" mitosis (involving complete NE disassembly), *Drosophila* embryos are an intermediate case, where the NE is partially disassembled, predominately at the poles (De Souza & Osmani, 2007; Katsani et al, 2008; Strunov et al, 2018). Despite the residual lamin localization as tubular-like structures, the NE is no longer intact after nuclear envelope breakdown (NEBD), and ER is the main continuous membranous structure surrounding the spindle (Fig S1). As the nuclear envelope broke down, we observed a higher concentration of ER in the vicinity of the spindle (Fig 1A, metaphase). The signal of Rtnl1–GFP also increased, exclusively around the spindle and most prominently at the spindle poles (Fig 1B). This is consistent with prior reports showing an increase in ER proteins at spindle poles upon mitotic entry (Bobinnec et al, 2003; Diaz et al, 2019). However, it remains unclear how these changes in ER organization impart on ER morphology across different stages of spindle assembly. Of note, both ER reporters shown are excluded from the spindle and resemble an envelope (Fig 1A and B, insets). At NEBD, we measured a reduction in the ER exclusion area (Fig 1C) supporting a transient contraction of the ER at this stage. This reduction is not accompanied by a decrease in the perimeter of the ER envelope (Fig 1E) but because of an indentation of the ER envelope at spindle poles (Fig 1A and B, arrows). At metaphase, we found that ER forms a large envelope around the spindle connected to two additional membranous structures surrounding each individual pole (Fig S2), similar to what has been recently reported in the first division of *Caenorhabditis elegans* embryos (Maheshwari et al, 2022 Preprint). During metaphase and anaphase, the area and perimeter of the ER exclusion gradually increase (Fig 1C and E), matching closely that of the spindle main body, disregarding the spindle poles (Fig 1D and F). This comparative measurement emphasizes the shape similarity of the ER envelope and the spindle body. At telophase, the ER undergoes shape changes that accompanied nuclear envelope reformation (Fig 1A and B, telophase and Fig S1). We also observed that both ER reporter proteins localized at the spindle midbody as previously reported (Bobinnec et al, 2003), suggesting that a considerable membrane reorganization occurs at this site (Fig 1A and B, telophase, arrowhead).

### ER reorganization throughout mitosis

Next, we wanted to understand if the observed ER shape changes, which closely follow those of the spindle body, arise from dynamic changes of ER membranes themselves. To this end, we performed FRAP experiments using the GFP-tagged transmembrane protein Rtnl1, which labels membranes that are being reshaped and tubular ER is being formed (Espadas et al, 2019). To circumvent the inherent changes in intensity and morphology during the cell cycle, which would impede proper FRAP analysis in these fast cycles, we have performed all the experiments in artificially arrested embryos. We first monitored the ER dynamics in interphase, by preventing mitotic entry with ectopic addition of the Cdk inhibitor p27 (Oliveira et al, 2010). Upon p27 injection, we observed that the ER maintained an interphase-like localization (Fig 2A). For FRAP studies, we bleached two different arrested nuclei in distinct regions (regions of interests [ROIs]) and imaged their fluorescence recovery over time (Fig 2A, red circles and Video 2). We observed a high turnover of ER membranes, with half-times of recovery in the order of seconds. ER membranes localized proximal to the spindle poles are more dynamic compared with those localized at the equator (Fig 2A, $t_{1/2}$ poles: 11.4 ± 6.2 s; $t_{1/2}$ equator: 21.0 ± 8.6 s). However, the Rtnl1–GFP mobile fraction at the poles is lower compared with the virtually complete recovery at the equator in interphase-arrested embryos (poles: 0.81 ± 0.06; equator: 0.9 ± 0.08). This high mobile fraction stands in contrast with the large immobile fraction for Rtn1/Rtn4a in yeast and mammalian cells (Shibata et al, 2008), implying that Rtnl1 is not an intrinsically diffusible protein within the ER. Altogether, our results suggest that Rtnl1 is more diffusible in *Drosophila* early embryos, or more likely, it reflects a high level of reshaping of ER membranes in these embryos.

Changes in ER morphology are coupled to the cell cycle and are dependent on cyclin A activity during *Drosophila* embryonic nuclear divisions (Bergman et al, 2015). To address if this morphological reorganization is accompanied by a change in the dynamic behavior of ER membranes, we repeated the same analysis in embryos arrested in metaphase. For this, embryos were microinjected with UbcH10$^{C114S}$, a dominant-negative catalytically dead version of the E2 ubiquitin-conjugating enzyme necessary for anaphase onset (Oliveira et al, 2010). We bleached two different nuclei in distinct ROI and imaged their fluorescence recovery over time (Fig 2B, red circles). Signal recovery of the ER-shaping protein Rtnl1 was in the same order of magnitude (Fig 2B and Video 3). Upon metaphase arrest, we detected a slightly slower recovery of Rtnl1–GFP intensity at spindle poles compared with the equatorial region (poles: 14.4 ± 3.2 s; equator: 11.8 ± 2.1 s). Interestingly, we observed virtually complete recovery of intensity in both cases (mobile fractions at pole: 0.92 ± 0.06; equator: 0.95 ± 0.04). This signal dynamics observed is slightly lower but within the same magnitude of the turnover observed for the luminal ER reporter Lys–GFP–KDEL (Fig 2C, $t_{1/2}$: 3.81 ± 0.46 s, Video 4), which reflects ER shape changes occurring in the time

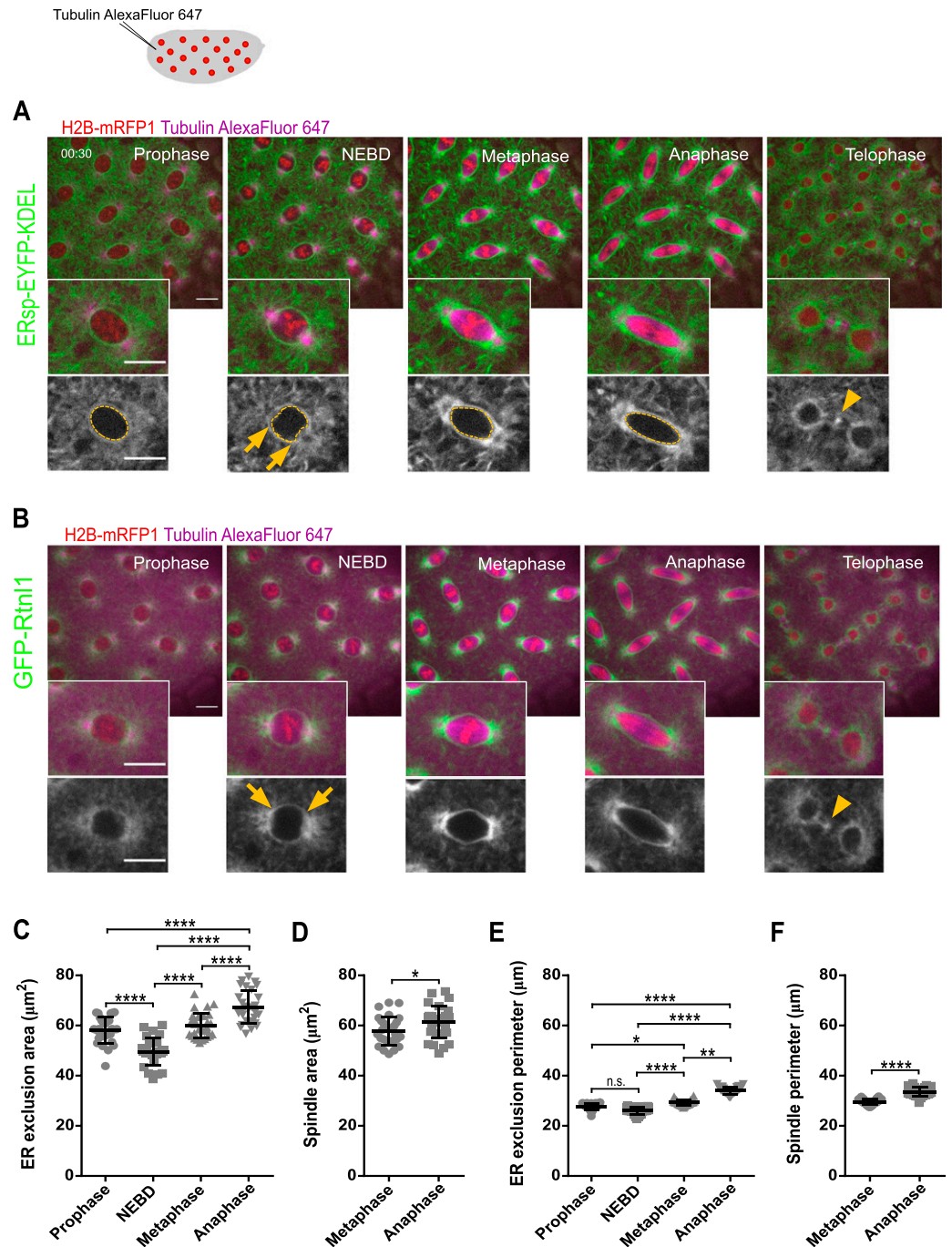

**Figure 1.  The ER forms an envelope surrounding the mitotic spindle in syncytial embryos.**
**(A, B)** Stills of embryonic divisions monitoring ER dynamics at different mitotic stages. ER was visualized with either EYFP-tagged ER retention sequence (ERsp-EYFP–KDEL, green, (A)) or the ER-shaping protein reticulon-like protein 1 (GFP–Rtnl1, green, (B)). Chromatin is labelled with Histone H2B–mRFP1 (red) and spindle microtubules with microinjected porcine tubulin labelled with Alexa Fluor 647 (magenta). Grey panels depict ER labelling alone. Scale bar is 10 $\mu$m. Arrowheads show events of ER abscission at telophase. **(C, D, E, F)** Quantifications of the ER exclusion area (C), spindle area (D), ER exclusion perimeter (E), and spindle perimeter (F), measured at the middle plane of the nuclei using the ERsp-EYFP–KDEL strain; sample size: N = 7 embryos per condition, n = 5 nuclei per embryo; statistical analysis was performed using one-way ANOVA, multiple comparisons, P value adjusted to multiple comparisons (Tukey) (C), Kruskal–Wallis (Dunn's multiple comparisons test) (E), or two-sided paired *t* test (D, F); *P < 0.05, **P < 0.01, ****P < 0.0001, n.s., nonsignificant, P > 0.05. Source data are available online for this figure.

scale of free diffusion events inside the ER. This is in agreement with previous studies, where rapid recovery of intensity was reported for both *Drosophila* oocyte fusome and syncytial embryos expressing Lys–GFP–KDEL (Snapp et al, 2004; Frescas et al, 2006). These findings suggest that the ER in mitosis is continuously undergoing significant reorganization.

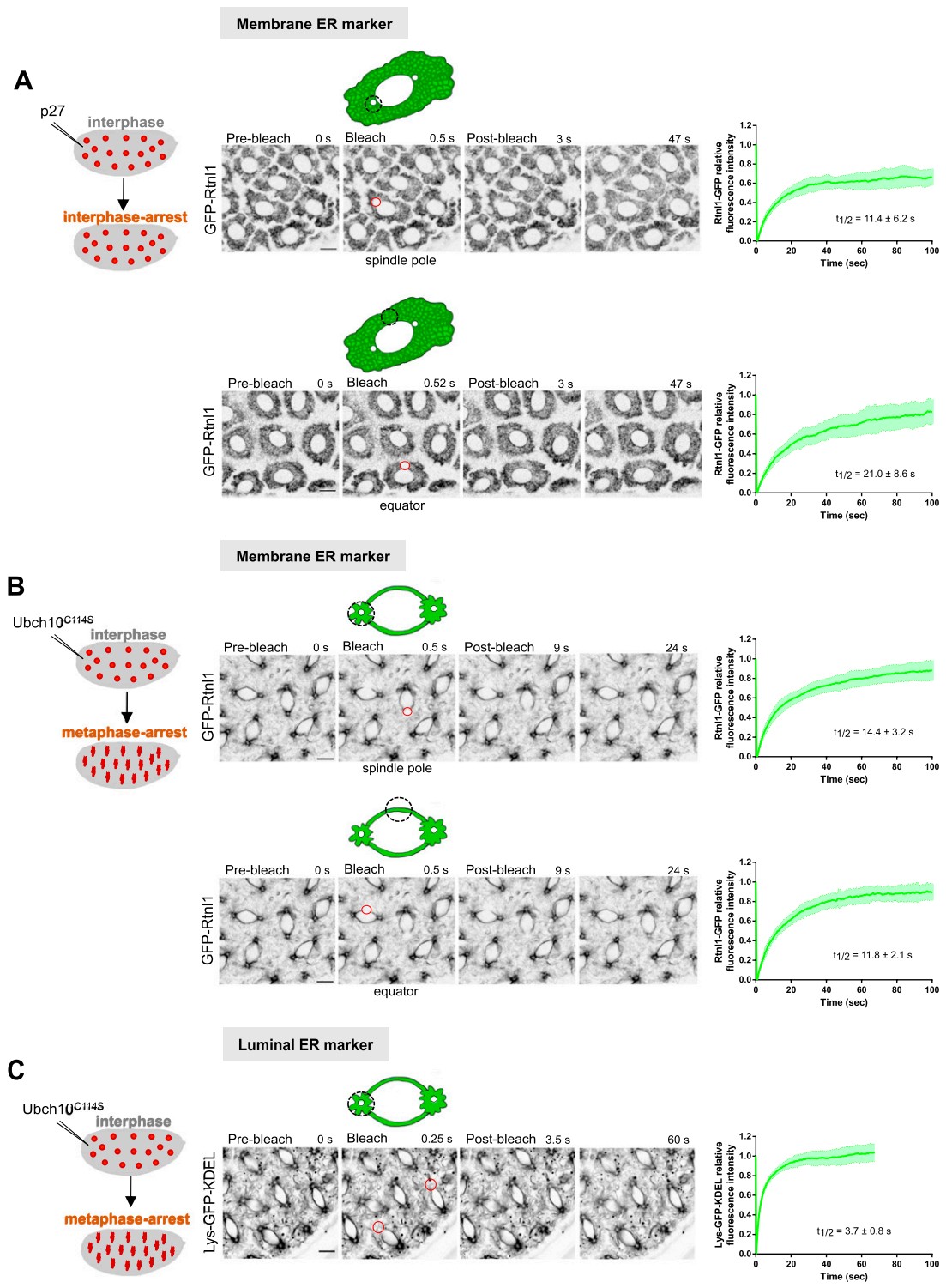

**Figure 2. ER dynamics in interphase- and metaphase-arrested nuclei.**
**(A)** FRAP of the membrane ER marker Rtnl1–GFP, in embryos microinjected with the Cdk inhibitor p27 to induce an interphase arrest. Recovery of Rtnl1–GFP fluorescence intensity was quantified upon bleaching of the ER at the spindle pole region (upper panel; N = 6 embryos, n = 12 nuclei) or at the equator (lower panel; N = 5 embryos, n = 9 nuclei). **(B)** FRAP analysis of Rtnl1–GFP in embryos microinjected with the dominant-negative version of the E2 ubiquitin-ligase UbcH10 (UbcH10$^{C114S}$) to induce a metaphase arrest. Graphs depict recovery of Rtnl1–GFP intensity upon bleaching at the spindle pole region (upper panel, N = 9 embryos, n = 17 nuclei) or at the equator (bottom panel, N = 9 embryos, n = 15 nuclei). **(C)** FRAP analysis of Lys–GFP–KDEL in metaphase-arrested embryos (UbcH10$^{C114S}$ injected). Graph depicts recovery of Lys–GFP–KDEL after photobleaching at spindle poles (N = 5 embryos, n = 10 nuclei). In all experiments, average half-times of recovery are depicted, presented as mean ± SD. Red circles depict the area bleached in each experimental condition. Scale bars are 10 μm. Source data are available online for this figure.

## Ectopic addition of cytATL changes mitotic ER topology at spindle poles

To explore how these dynamic ER membranes could impact on the overall architecture of mitosis, we developed a strategy to acutely perturb the ER in metaphase-arrested embryos and follow the consequences in real-time. To this end, we made use of the cytosolic domain of the ER membrane fusing protein atlastin (Hu et al, 2009; Bian et al, 2011). This truncated version has been used as a dominant-negative reagent in *Xenopus* egg extracts and human cells to impair the membrane fusion activity of the native form on the ER (Wang et al, 2013, 2016; Kutay et al, 2021). *Drosophila melanogaster* cytATL was expressed in and purified from bacteria (Fig S3) and microinjected into syncytial embryos. To monitor the ER specifically during metaphase, embryos were previously arrested with UbcH10$^{C114S}$, as above (Fig 3A). We observed that subsequent microinjection of cytATL in metaphase-arrested embryos alters mitotic ER topology compared with controls (Fig 3B and C and Video 5). Over time, ERsp-EYFP–KDEL–labelled ER membranes lose their linear arrangement, typical of a tubular structure, and acquire diffuse and homogeneous appearance (Fig 3B and C, t = 10:00 min). Quantitative analysis reveals that there is no change in the mean intensity of the ER reporter at spindle poles, with or without cytATL injection (Fig 3D). However, the spatial distribution of the signal altered, as evidenced by the significantly decreasing variance upon cytATL injection (Fig 3E). This suggests that the ER contents remain but their distribution or local concentration changes. In contrast, centrosome-distal regions do not change significantly both in mean and distribution of the signal (Fig 3D and E). These findings reveal that the disruptive effect of cytATL is exclusively observed at ER membranes surrounding spindle poles. This selective effect is consistent with the strong enrichment of atlastin-EGFP at spindle poles during mitosis (Fig S4). In addition to the change in spatial ER concentration, ectopic addition of cytATL caused a significant reduction in the ER exclusion zone (Fig 3F). Importantly, microinjection of a cytATL fragment mutated in the dimerization domain (cytATL$^{R192Q}$), unable to dominantly compete with endogenous ATL (Bian et al, 2011; Byrnes & Sondermann, 2011; Ulengin et al, 2015), does not cause the same degree of ER fragmentation nor a reduction in the exclusion zone (Fig S5A–D). We therefore concluded that ER membranes at the spindle poles are more sensitive to the disruptive effect of cytATL, via dimerization with endogenous ATL, whose addition leads to acute changes in morphology at this subregion of the ER network.

## Acute disruption of spindle pole-proximal ER membranes decreases mitotic spindle size

Having established a method that acutely disrupts ER integrity in mitosis, particularly at centrosome-proximal regions, we next sought to evaluate the effect this disruption has on mitotic spindle architecture. For this, we co-injected UbcH10$^{C114S}$ with porcine tubulin labelled with Alexa Fluor 647 to visualize spindle MTs during the metaphase arrest (Fig 4A and Video 6). Upon cytATL-induced ER disruption, in contrast to controls, we found that the morphology of the spindle is significantly altered (Fig 4B and C, t = 10:00 min). These changes in spindle architecture are evidenced by a reduction in

spindle length and width (Fig 4D and E). In contrast, microinjection of mutant cytATL$^{R192Q}$ does not alter spindle size (Fig S5E and F). Thus, we conclude that ER morphological changes mediated by the dominant-negative effect of cytATL lead to smaller mitotic spindle sizes. In addition to this effect, we also observed, at considerable frequency, the detachment of the spindle pole MT-organizing center (MTOC) upon ectopic addition of cytATL. To quantify this, we measured the distance between the focal point of the spindle body and the MTOC; this distance was markedly higher upon ectopic addition of cytATL, on average three times the distance measured in control embryos (Fig 4F, 0.4 ± 0.2 μm versus 1.2 ± 0.5 μm). We also observed that upon perturbation of centrosome-proximal ER membranes, there was a more frequent split of the centrosomes, corresponding to mother/daughter centriole disengagement (Fig S6). We measured the distance between two adjacent poles identified in tubulin-labelled spindles, in cases where a split could be identified (otherwise defined as zero: no disengagement). The frequency of long distances increased upon cytATL addition (Fig 4G). This observation suggests a role for the ER in centrosome dynamics. Overall, our spatially resolved perturbation of ER membranes reveals that the ER surrounding the spindle poles plays multiple roles in spindle architecture, including spindle size, spindle pole attachment, and centrosome engagement.

Given the observed changes in spindle architecture upon impairment of spindle pole-proximal ER membranes, we next investigated whether spindle dynamics would also be affected. We used FRAP to bleach one half of the spindle in metaphase-arrested embryos and analyzed spindle MT turnover. We used embryos expressing β-tubulin–GFP and monitored the recovery of the fluorescence intensity after photobleaching (Fig 4H and Video 7). In control (buffer-injected) conditions, this analysis revealed that spindle MTs recover very fast, with a half-time of recovery of 10.7 ± 2.2 s (Fig 4H right, grey trace). Such fast and full recovery upon bleaching of half the spindle is consistent with previous studies (Brust-Mascher et al, 2004) and reflects the high abundance of free tubulin (maternal load) (Raff et al, 1982) in combination with high spindle MT turnover.

We next repeated the same analysis after cytATL-mediated ER disruption. In this condition, we found that MT turnover remained unaltered relative to controls (Fig 4H, magenta trace). These results imply that despite the marked difference in spindle size, the dynamic behavior of MTs remains unaltered. To confirm this notion, we measured MT growth using the plus-end protein EB1 and time-lapse imaged embryos expressing a GFP-tagged EB1 protein (EB1–GFP) in control or cytATL-injected embryos (Fig 4I, left and Video 8). Because EB1–GFP localizes to growing MT ends, it displayed moving speckles within the spindle in time-lapse images. To estimate MT growth speed, we generated space-time projections (kymographs) along the spindle axis, which transform the continuously moving speckles into linear signals (Fig 4I, middle). Analysis of growth speed revealed no significant differences between control and cytATL-injected embryos (Fig 4I, right). Thus, we conclude that the observed changes in spindle architecture upon ectopic addition of cytATL are not accompanied by changes of MT dynamics. From this insight, we hypothesized that the ER confers a constraining mechanical force on the mitotic spindle, which could play a role in spindle function.

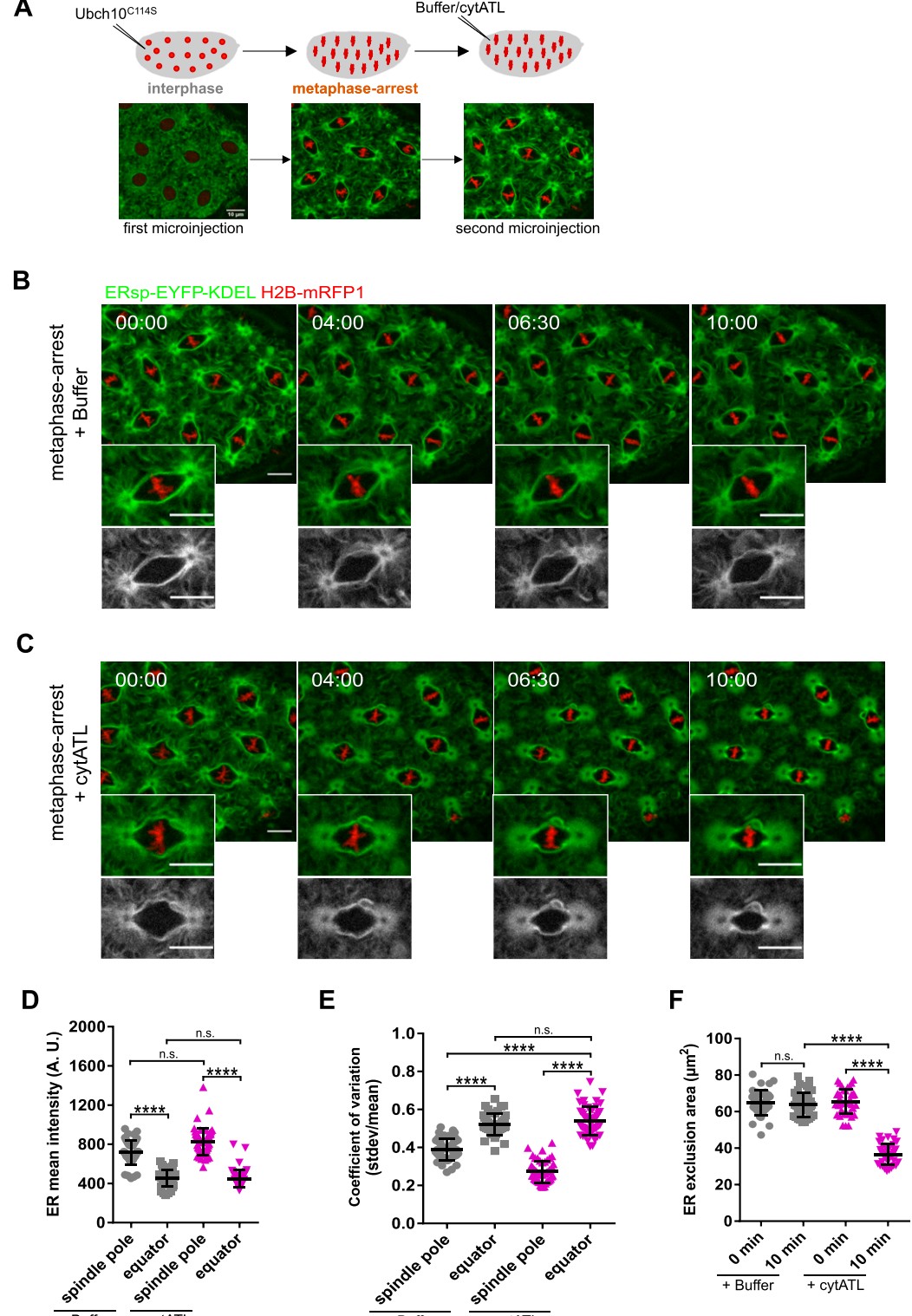

**Figure 3. Acute ER disruption is achieved by ectopic addition of cytATL.**
**(A)** Experimental layout for acute ER disruption: embryos were arrested in metaphase by microinjection with UbcH10$^{C114S}$ and allowed to reach chromosome alignment. After arrest (~5 min) embryos were subjected to a second microinjection. ER dynamics was monitored by ERsp-EYFP–KDEL (green) and chromatin by H2B–mRFP1 (red). **(B, C)** Stills depicting changes in ER morphology (ERsp-EYFP–KDEL, green) after microinjection of buffer (B) or the dominant-negative cytATL protein (C) in metaphase-arrested embryos; time (min:s) is relative to the second microinjection (buffer/cytATL). Grey panels depict ER channel alone. Scale bar is 10 $\mu$m. **(D)** Quantitative analysis of the mean fluorescence intensity of ERsp-EYFP–KDEL in spindle poles or equatorial regions 10 min after buffer/cytATL injection. **(E)** Coefficient of variation, that is, the

## cytATL-mediated disruption of the ER impairs mitotic spindle pulling forces

We next asked how the smaller spindles that result from acute and local perturbation of spindle pole-proximal ER membranes behave at the functional level. For this, we took advantage of a molecular tool that enables the artificial separation of sister chromatids. With this approach, fast cohesin inactivation is achieved by the tobacco etch virus (TEV) protease in embryos surviving on a modified version of Rad21 that contains TEV cleavage sites (Pauli et al, 2008). Upon microinjection of TEV protease, sister chromatid separation is observed within 1–2 min (Oliveira et al, 2010; Carmo et al, 2019). This tool allowed us to investigate changes in pulling force on chromatids, as poleward movement of separated sister chromatids relies on how efficiently they are pulled apart by spindle MTs. Embryos were first arrested in metaphase, followed by the ectopic addition of cytATL or buffer. 10 min after cytATL/buffer injection, embryos were injected with TEV protease to trigger artificial sister chromatid separation (Fig 5A and Video 9). Under both conditions, sister chromatids separated (Fig 5B and C). Analysis of chromosome movement was performed based on kymographs that display chromosome position over time (Fig 5B and C, right) and area occupied by chromosomes (Fig S7A). In control embryos, we found that sister chromatid separation, defined by the bifurcation in the kymograph (Fig 5B and C right, arrows), is elicited within ~2 min upon microinjection with TEV protease, whereas it is delayed upon ectopic addition of cytATL (Fig 5D). Moreover, the velocity of initial poleward-movement of sister chromatids along the spindle axis, as estimated by the angle of signal bifurcation in the kymograph, is reduced upon cytATL-driven ER disruption compared with the control condition (Fig 5E). The range of chromatid separation along the spindle axis is also reduced upon cytATL addition (Figs 5F and S7B), which we attribute to lower separation velocity and an overall shorter spindle size. After the initial separation, isolated sisters from both experimental conditions (buffer/cytATL) were equally able to engage into oscillatory movements driven by cycles of chromosome capture/detachment (Fig S7C). We conclude that the short spindles imposed by acute ER disruption are still able to pull and capture chromatids. However, poleward chromatid movement occurs at a slower velocity, implying a decrease in pulling forces on chromatids.

## cytATL-driven disruption of the ER affects nuclear spacing upon release from the metaphase arrest

Our previous work has revealed that the spindle pole MTOC plays a crucial role in daughter nuclei separation and nuclear spacing in the syncytial embryo (Telley et al, 2012; de-Carvalho et al, 2022). Having observed spindle pole detachment upon cyATL injection (Fig 4), we next aimed to probe nuclear separation after disruption of the ER in embryos undergoing the changes characteristic of a normal mitotic exit. For this we took advantage of an inducible release of the metaphase arrest reported previously (Piskadlo et al, 2017): we induced a metaphase arrest in division cycle 10 using the dominant-negative UbcH10$^{C114S}$ for 5 min, followed by injection of buffer/cytATL, and waited for 10 min to allow for ER disruption. Embryos were subsequently injected with a WT version of UbcH10 protein, which induces anaphase onset and, thus, mitotic exit in 4–8 min (Fig 5A, G, and H and Video 10). Using this approach coupled to time-lapse imaging, we then generated kymographs of the chromatin signal and tracked chromosome segregation and daughter nuclei separation (Fig 5G and H, right). This analysis revealed that sister chromatids are separated at a similar velocity during anaphase in both control and cytATL conditions (Fig 5I). This contrasts with what we observed in embryos arrested in metaphase (+TEV cleavage of cohesin), suggesting that anaphase-specific changes may compensate for the reduced pulling forces observed upon ER disruption in metaphase. However, despite normal segregation speeds (Fig 5I), daughter nuclei were not efficiently separated during telophase and early interphase upon ER disruption (Fig 5J). This process is driven by the centrosome-nucleated MT aster (Telley et al, 2012) and, therefore, the reduced separation upon ER disruption is likely caused by the defects observed on spindle pole attachment. Furthermore, the inefficient separation led to a lower nuclear ordering in the subsequent interphase. In our control experiments, the UbcH10-arrest/release approach is able to reproduce the non-sibling internuclear distance recently reported for this division cycle (cycle 11) (de-Carvalho et al, 2022). In contrast, upon ER disruption, sibling nuclear distance was shorter and the distances between non-sibling nuclei were longer when compared with the control condition (Fig 5K and L).

In summary, here we show that ER membranes located in the vicinity of the spindle poles are critical for maintenance of proper mitotic spindle shape and function. This novel role for ER membranes is required throughout metaphase, even after unperturbed spindle assembly. Our findings highlight that the role of the ER in spindle architecture goes well beyond the phase of spindle assembly at early mitotic stages, as previously reported (Liu & Zheng, 2009; Schweizer et al, 2015). We favor that this constant requirement may underlie the observed continuous remodeling of ER membranes, as evidenced by the highly dynamic behaviors of mitotic ER membranes. Cdk1 consensus sequences were found in *Drosophila* ER-shaping proteins such as Rtnl1, spastin, and atlastin (Bergman et al, 2015), suggesting that their activity is spatiotemporally regulated throughout the cell cycle. Moreover, the human ortholog of atlastin-1 interacts with spastin, a MT-severing ATPase, within tubular ER membranes in neurons (Park et al, 2010), molecularly linking ER shaping and MT dynamics. However, it remains to be determined how the centrosome-proximal ER membranes maintain spindle shape in dividing tissues. Previous work suggested that

---

ratio of stdev over the mean, 10 min after buffer/cytATL injection. **(F)** ER exclusion area in control (buffer injected, grey) and cytATL (magenta) conditions at the first (t = 0 min) and last (t = 10 min) time point of the time-lapse. Statistical analysis using N = 10 embryos, n = 5 nuclei per embryo. **(D, E, F)** Asterisks represent statistical significance derived from Kruskal–Wallis (Dunn's multiple comparisons test) (D) or one-way ANOVA (E, F), multiple comparisons, *P* value adjusted to multiple comparisons (Tukey). ****$P$ < 0.0001, n.s., nonsignificant, $P$ > 0.05.
Source data are available online for this figure.

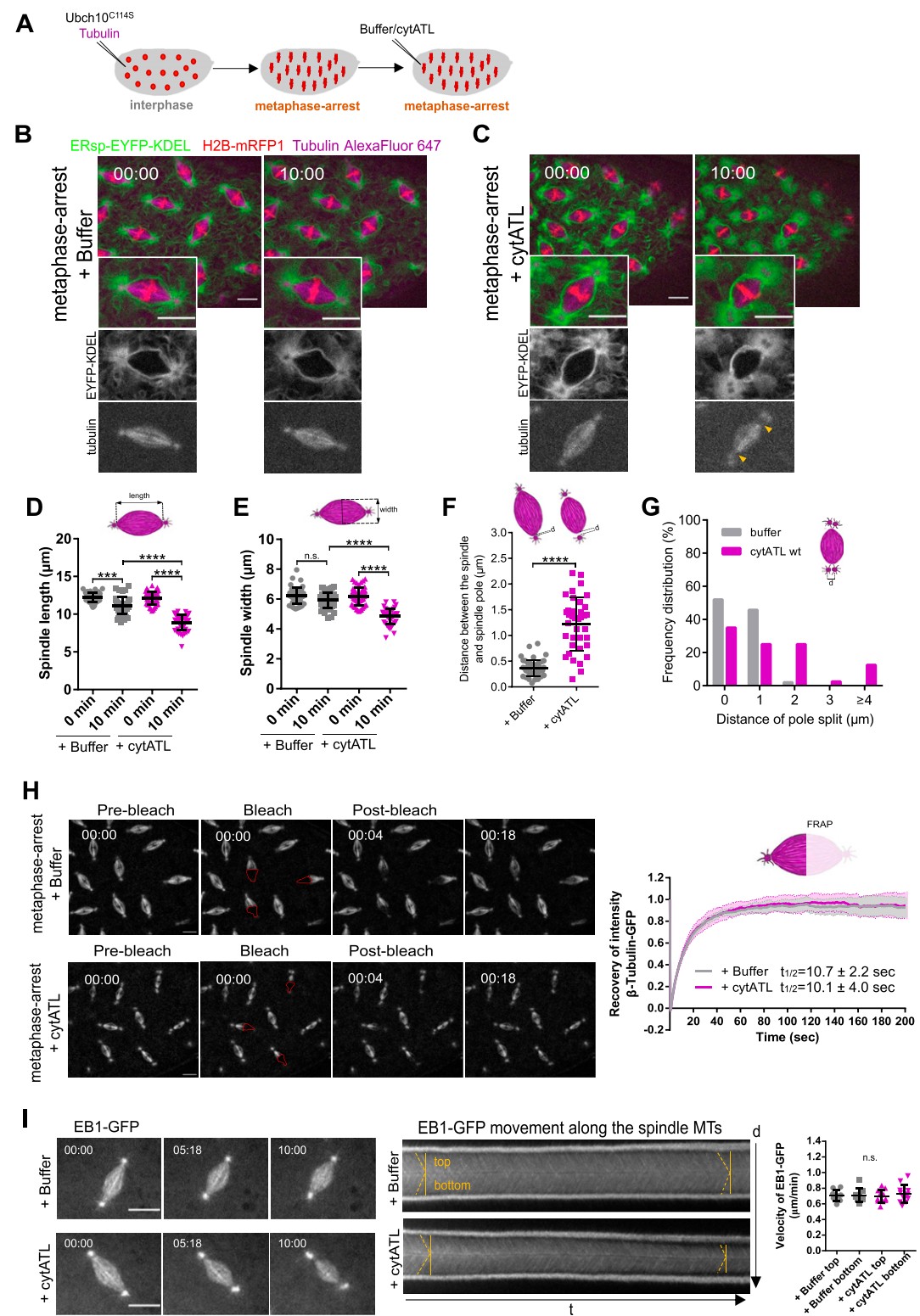

**Figure 4. Spindle shape and function are affected upon cytATL-mediated ER disruption.**
**(A)** Schematics of the experimental layout: embryos were microinjected with UbcH10[C114S] and Alexa Fluor 647–labelled tubulin (to visualize spindle microtubules), followed by subsequent microinjection with buffer or cytATL. **(B, C)** Representative control (B) and cytATL-injected embryos (C), showing the ER (ERsp-EYFP–KDEL, green), chromosomes (H2B–mRFP, red), and the spindle (magenta) at the first (t = 00:00 min) and last (t = 10:00 min) time points after microinjection with buffer/cytATL. Grey panels depict ER (top) and spindle (bottom) alone. Arrowheads highlight spindle pole detachment observed upon microinjection with cytATL. Scale bar is 10 μm.
**(D, E)** Quantification of spindle length (D) and width (E) in control (+buffer, grey) and cytATL (+cytATL, magenta) embryos at the first (t = 0 min) and last (t = 10 min) time

membranous structures surrounding the spindle can affect molecular crowding and thereby impact spindle assembly (Schweizer et al, 2015). Although we cannot exclude whether changes in the molecular composition are also imposed in our experiments, the finding that spindle dynamics remains largely unaltered strongly suggests this is not the case. Instead, we favor that ER membranes at the spindle poles may aid in the balance of forces within the spindle. Mechanical force transmission and balancing may be related to the attachment of the spindle pole MTOC. A role for the ER in centrosome dynamics and function has been highlighted in a recent study (Maheshwari et al, 2022 Preprint), which revealed that membrane poles surround the centrosomes—termed centriculum—during the first mitotic division in *C. elegans*. These ER membranes impact centrosome properties, spindle assembly, and nuclear envelope breakdown. Anchoring of the centrosome and MT aster to the nuclear envelope is particularly relevant during syncytial divisions, as ordered nuclear positioning depends on MT asters (Telley et al, 2012; de-Carvalho et al, 2022). ER may also have a prominent role in compartmentalization in this multinucleated system, further adding robustness to nuclei positioning. The spatial distribution of dividing nuclei is a crucial event for later developmental steps, such as cell size determination and gene expression patterning. Future work should address the role of the ER in other cells where nuclear positioning is critical, including those with canonical anchoring patterns, with direct implications on further organism development.

# Materials and Methods

### Fly stocks

Fly strains expressing UASp-Lys–GFP–KDEL (Snapp et al, 2004), ERsp–EYFP–KDEL (LaJeunesse et al, 2004), Rtnl1–GFP (Morin et al, 2001) fluorescent markers, and for induction of artificial sister chromatid separation (Pauli et al, 2008; Oliveira et al, 2010) have been previously described. To produce flies expressing endogenous EGFP-tagged atlastin, we have used the Fly transgenics service at the Instituto Gulbenkian de Ciência, using a CRISPR/Cas9-mediated genome editing approach. A list with all stocks can be found in Table 1.

### Generation of atlastin-GFP knock-in using CRISPR/Cas9

The knock-in fly line was generated by the Fly Transgenesis and Genome Editing Facility. To visualize the localization of endogenous atlastin in *Drosophila* syncytial embryos, a knock-in of the EGFP sequence into the C-terminal region of the atlastin locus was generated. The following guide RNAs were used according to a previous study (Port et al, 2014): first guide RNA–selected target: GTGGGAGAAAGTAAGTAGCTGGG and second guide RNA–selected target: GCCATTGGACGCATTCACCGAGG. The EGFP sequence flanked by ~1 kb homology region was used as a donor. The stock BL#66554: y[1] M{RFP[3xP3.PB] GFP[E.3xP3]=vas-Cas9}ZH-2A was injected in the germline. Founders were crossed with the w; ; MKRS/TM6B stock, and 115 lines were established. A positive knock-in was selected by PCR and confirmed by sequencing.

### Microinjections

Microinjections were performed as previously described in Carmo et al (2019), using an Eppendorf FemtoJet Microinjector controller and commercially available needles (Eppendorf Femtotips Diameter [Metric] Inner: 0.5 $\mu$m, 11883991). Briefly, dechorionated embryos (1–1.5 h old) were aligned and glued onto a #1.5 coverslip (24 × 40 mm), dried at room temperature for 13 min, and subsequently covered in halocarbon oil.

Buffer/protein/chemicals were microinjected at the following concentrations: buffer (10 mM Hepes, 100 mM KCl, 1 mM $MgCl_2$, 10% glycerol), UbcH10$^{C114S}$ (30–40 mg/ml), p27 (4 mg/ml), TEV protease (18 mg/ml), cytATL (60 mg/ml), and UbcH10 WT (60 mg/ml). Porcine tubulin labelled with Alexa Fluor 647 (Tubulin HILyte Fluor 647 labelled, cat. # 1L670M, 20 $\mu$g; Cytoskeleton) was reconstituted in freshly prepared and filtered 1× BRB80 buffer (80 mM PIPES, 1 mM $MgCl_2$, 1 mM EGTA, pH 7.8) to a final concentration of 2.5 mg/ml, immediately flash frozen in liquid nitrogen, and stored at –80°C. To visualize spindle MTs upon metaphase arrest, labelled tubulin was mixed with UbcH10$^{C114S}$ protein at 1:1 ratio (at a final concentration of 1.25 mg/ml). To visualize spindle MTs during nuclear division cycles, tubulin was used at 2.5 mg/ml.

### Cloning of dimerization mutant cytATL$^{R192Q}$

The plasmid containing the sequence of *Drosophila* cytATL (residues 1–422), pET28a-His$_6$-cytATL, was a gift from Jiunjie Hu (Nankai University, China). To generate the dimerization mutant cytATL-R192Q, we used a site-directed mutagenesis approach with the primers forward – 5′-agcgcctgcagttcctggttcaggattggagcttcccgtatga-3′ and reverse – 5′-tcatacgggaagctccaatcctgaaccaggaactgcaggcgct-3′. Phusion (HF) DNA polymerase was used with a modified PCR reaction (50 $\mu$l volume) optimized for this purpose adding the following reagents: 5× Phusion HF buffer, forward primer at 10 $\mu$M,

points of the time lapse. Asterisks depict the statistical significance derived Kruskal–Wallis test (D, E), Dunn's multiple comparisons test ***$P$ = 0.0001, ****$P$ < 0.0001, n.s. $P$ > 0.05 (N = 10 embryos, n = 5 nuclei). **(F)** Distance d between the focal point of the spindle and the spindle pole. Asterisks depict the statistical significance derived from the unpaired two-tailed Mann–Whitney (nonparametric) test, ****$P$ < 0.0001 (N = 10 embryos, n = 5 nuclei). **(G)** Frequency distribution of spindle pole split (distance, d) in control (+buffer, grey) and cytATL (+cytATL, magenta) at the last time points of the time lapse (t = 10 min). **(H)** FRAP assay of $\beta$-tubulin–GFP in UbcH10$^{C114S}$-arrested embryos (metaphase arrest) in control (buffer injection) and ER disruption (+cytATL) conditions. Bleached areas are marked in red. Right: Quantification of recovery of $\beta$-tubulin-GFP intensity after bleaching in both conditions, with calculated half times of recovery (mean ± SD, N = 5 embryos, n = 3 nuclei). **(I)** Analysis of MT growth rate with and without ER disruption (buffer/cytATL injection) in metaphase arrested embryos. Microtubule plus-ends were monitored using a GFP-tagged EB1 transgene. Time (min:s) is relative to microinjection with buffer/cytATL. Scale bar is 10 $\mu$m. Middle: Kymographs of EB1–GFP intensity were generated, and the angles of EB1 tracks (yellow lines) were used to estimate the velocities of MT growth, shown on the right. Statistical analysis was performed using one-way ANOVA. n.s., nonsignificant, $P$ > 0.05 (N = 4 embryos, n = 3 nuclei).
Source data are available online for this figure.

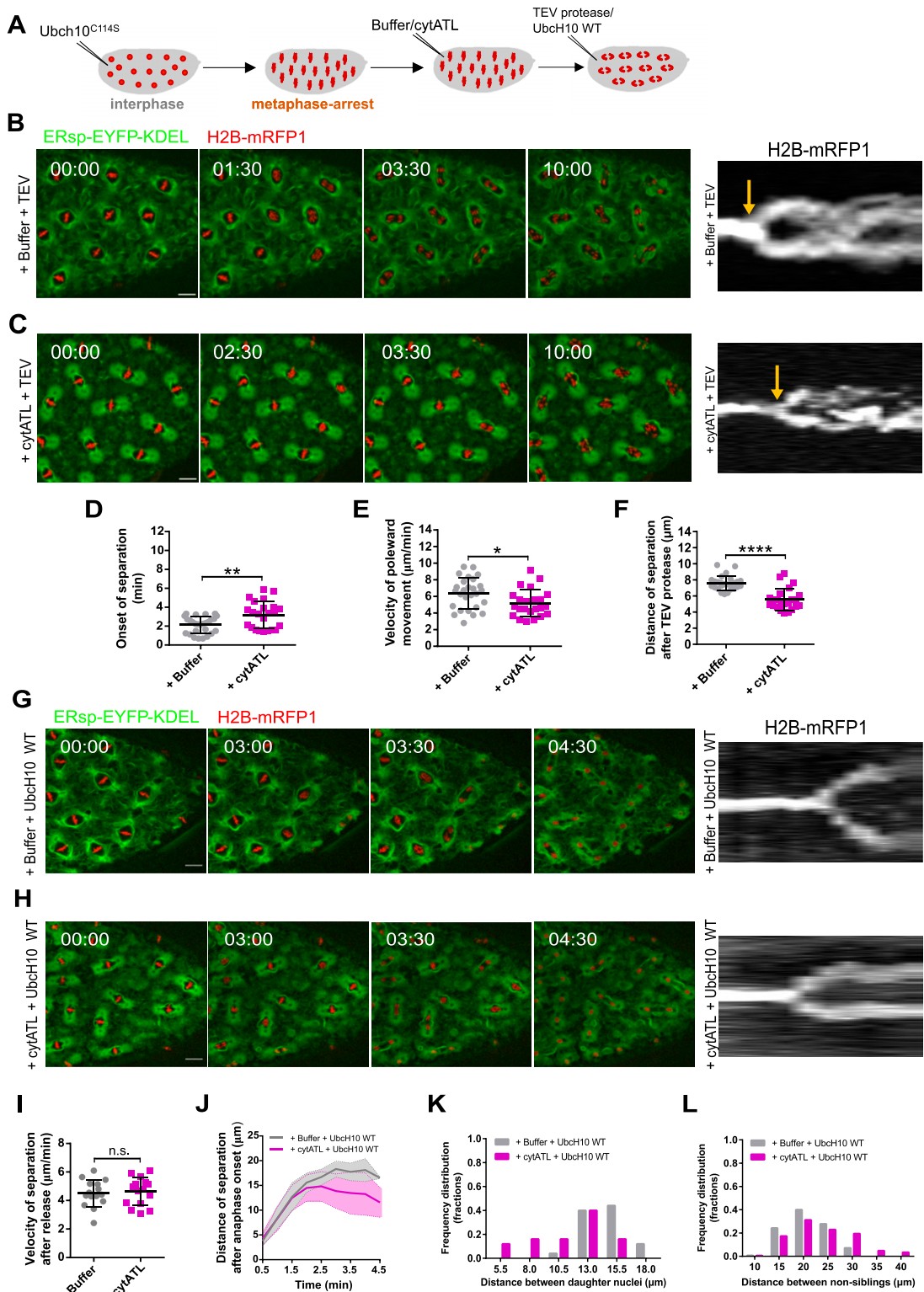

**Figure 5. Spindle function is impaired upon cytATL-mediated ER disruption.**

**(A)** Experimental layout: embryos were arrested in metaphase (UbcH10$^{C114S}$ microinjection), followed by microinjection with buffer/cytATL. After 10 min, embryos were subjected to a third microinjection with either TEV protease, to induce acute sister chromatid separation, or UbcH10$^{WT}$ to trigger mitotic exit. Embryos surviving solely on a TEV-cleavable version of Rad21 (Cohesin) were used for these experiments. **(B, C)** Time-lapse images from embryos after microinjection with buffer/cytATL and subsequently microinjected with TEV protease to trigger cohesin cleavage (+buffer+TEV, B; +cytATL+TEV, C); ER is labelled in green (ERsp–EYFP–KDEL) and chromosomes in red (H2B–mRFP1); scale bar is 10 $\mu$m; time (min:s) is relative to microinjection with TEV; kymographs (right panel) depict chromosome positioning over time; arrows

reverse primer at 10 $\mu$M, 10 mM dNTPs, 3% DMSO, 3.75 $\mu$l of MgCl$_2$ at 25 mM, template DNA at 10 ng/$\mu$l, 0.5 $\mu$l Phusion HF, and nuclease-free water. The following PCR protocol was used: initial denaturation at 95°C for 5 min, then denaturation at 95°C for 50 s, annealing at 72°C for 50 s, extension at 72°C for 10 min for 17 cycles, final extension at 72°C for 30 min, and holding at 12°C. The entire reaction volume was used to run an agarose gel electrophoresis. The band corresponding to the size of the pET28a-His6-cytATL-R192Q plasmid (~6,000 bp) was cut using a scalpel, and DNA was purified using the Zymoclean Gel DNA Recovery Kit according to manufacturer instructions. The resulting nicked plasmid was transformed into chemically competent *E. coli* DH5$\alpha$ cells cultured with the appropriate selective conditions (kanamycin resistance), and a miniprep was performed to extract the DNA (QIAGEN kit). The plasmid was sent for sequencing to confirm the modified residue. For Sanger sequencing (performed by Eurofins genomics), we used the two primers described above, T7 promoter, T7 terminator, and the reverse 5'-GCACTCGAGgctttcgttgtgcgcctgg-3'.

### Protein purification

UbcH10$^{C114S}$, p27, and TEV protease were purified as previously described (Oliveira et al, 2010; Piskadlo et al, 2017). For cytATL, the bacterial expression plasmid containing *D. melanogaster* cytATL sequence (pET28a-His$_6$-cytATL; kindly provided by Junjie Hu, Nankai University, China) was used to transform *E. coli* BL21 cells. Bacterial cells from an overnight-grown starting culture were grown in 1 liter LB medium, supplemented with kanamycin (50 $\mu$g/ml), and incubated at 37°C. Protein expression was induced by adding 0.2 mM IPTG once the O.D.$_{600}$ was at 0.8. The culture was grown at 18°C overnight, and bacteria were harvested by centrifugation at 3,374*g* for 45 min, at 4°C. For protein purification, the following buffers were prepared freshly and filtered: 0.1 M KPi at pH 7.2; wash buffer (50 mM KPi pH 7.2, 400 mM NaCl, 1 mM ß-ME, 7 mM imidazole), lysis buffer (50 mM KPi, pH 7.2, 400 mM NaCl, 1 mM ß-ME, 7 mM imidazole, 0.1% Triton X-100 supplemented with one tablet of protease inhibitors (Pierce Protease Inhibitor Mini Tablets, EDTA-free, reference A32955; Thermo Fisher Scientific) and DNAseI), elution buffer 1 (50 mM KPi, pH 7.2, 400 mM NaCl, 1 mM ß-ME, 400 mM imidazole), elution buffer 2 (50 mM KPi, pH 7.2, 400 mM NaCl, 1 mM ß-ME, without imidazole). Bacterial pellets were resuspended in 10 ml lysis buffer, adding DNase and a protease inhibitor tablet onto the pellet directly. The lysate was then passed through a French press system (Emulsiflex C5 High Pressure Homogenizer). Protein extract was spun at 18,800*g* for 45 min at 4°C, and the pellet was discarded. Ni sepharose beads (2–3 ml; GE Healthcare) were added into a 50-ml

Falcon tube. Protein extract was incubated with beads for 1 h at 4°C with gentle agitation. Beads were washed three times with wash buffer (each round at 700 *g* for 5 min) and packed into a COLUMN PD-10 EMPTY (GE Healthcare). The protein was eluted in distinct fractions with increasing concentrations of imidazole (80, 150, 300, 400 mM). The two fractions eluted with 300 and 400 mM imidazole, respectively, were pooled together and dialyzed overnight at 4°C using a cassette (Slide-A-Lyzer Dialysis Cassettes, 7 KDa MWCO, 12 ml; Life Technologies) into a 1 liter volume of cytATL storage buffer (10 mM Hepes, 100 mM KCl, 1 mM MgCl$_2$, 10% glycerol). To concentrate the purified cytATL protein, we used Amicon ultra-centrifugal filter 15 ml with a 10 kDa cut-off (Millipore) and protein concentration was quantified in a nanodrop using the A280.

For cytATL$^{R192Q}$ purification, we used the same protocol for cytATL WT with some modifications to maximize protein stability. 1 mM MgCl$_2$ was added to the wash, lysis, and elution buffers. The protein extract was incubated with Talon metal affinity resin (635502; TaKaRa) for 1 h at 4°C with gentle agitation. The protein was then eluted using 20, 80, and 150 mM imidazole. The 150 mM imidazole fraction was buffer exchanged immediately into storage buffer (10 mM Hepes, 100 mM KCl, 1 mM MgCl$_2$, 10% glycerol) using a PD-10 Sephadex G-25 M column (17-0851-01; GE Healthcare). To concentrate the purified cytATL$^{R192Q}$ protein, we used Amicon ultra-centrifugal filter 15 ml with a 10 kDa cut-off (Millipore) and protein concentration was quantified in a nanodrop using the A280.

### Microscopy

Time-lapse movies of live embryos were obtained using Confocal Z-series stacks with a Yokogawa CSU-X Spinning Disk confocal, mounted on a Leica DMi8 microscope, with a 63× 1.3 NA glycerine immersion objective, using the 488- and 561-nm laser lines and a Andor iXon Ultra EMCCD 1024x1024 camera. The system was controlled with Metamorph software (Molecular Devices). For the sequential microinjection experiments, we used 30-s time points and a total of 10 min for each time-lapse acquisition, with 0.4–0.5 $\mu$m z-step size and 15 slices. For 3D rendering of nuclei, we imaged a z-stack in Nyquist sampling in z (10 $\mu$m, 0.22 $\mu$m z-step size, 51 slices).

### Quantitative imaging analysis

To quantify the area and perimeter of the ER exclusion zone and of the spindle, a single z slice corresponding to the middle plane of each nucleus was used. All the measurements were performed using the segmented line tool in Fiji (yellow shapes in Fig 1A, insets), for each channel (ER, spindle) separately.

---

highlight the onset of chromatid separation. **(D, E, F)** Quantification of the onset of separation (D), velocity of poleward movement (E), and distance of chromatid separation (F) induced by cohesin cleavage (+TEV), in conditions with intact ER (+buffer) or disrupted ER (+cytATL). Sample size: N = 5 embryos, n = 5 different nuclei for each experimental condition. **(G, H)** Time lapse images of induced anaphase in unperturbed ER (G, +buffer+UbcH10$^{WT}$) and perturbed ER (H, +cytATL+UbcH10$^{WT}$) conditions; ER is labelled in green (ERsp-EYFP–KDEL) and chromosomes in red (H2B–mRFP1); scale bar is 10 $\mu$m; time (min:s) is relative to microinjection with UbcH10$^{WT}$; kymographs (right panel) depict chromosome positioning over time; **(I, J)** Quantification of the velocity of chromosome separation (H) and distance of chromosome separation (J) triggered upon artificial induction of anaphase (UbcH10$^{WT}$) in embryos previously injected with buffer or cytATL. Sample size: N = 5 embryos, n = 3 different nuclei for each experimental condition. **(K, L)** Frequency distributions of distance between daughter nuclei (K) and between non-sibling nuclei (L) measured 3 min after anaphase onset; sample size: three to four measurements in N = 5 embryos **(K)** and six measurements in N = 5 embryos (L). **(D, E, F)** Asterisks refer to statistical significance, derived from unpaired *t* tests (two-sided) (D, E), nonparametric Mann–Whitney test (F) *$P < 0.05$, **$P < 0.01$, ***$P < 0.001$, ****$P < 0.0001$, n.s., nonsignificant, $P > 0.05$. Source data are available online for this figure.

**Table 1.  List of *Drosophila* stocks used in this study.**

| Genotype | Reference |
|---|---|
| *Avic\GFP*<sup>EYFP.sqh.Tag:SS(hCALR).Tag:ER(KDEL)</sup> *(ERsp-EYFP–KDEL)* | BDSC #7195. LaJeunesse et al (2004) |
| *w\*; P{w*<sup>+mC</sup>*=PTT-un1}G00071 GFP*-tagged *Rtnl1* protein expressed from its endogenous locus *(GFP–Rtln1)* | Kyoto stock center #110579. Morin et al (2001) |
| *w[\*]; P{w[+mC]=UASp-Ggal\LYZ.GFP.KDEL}401/CyO* Expresses *GFP*-tagged chicken lysozyme with an ER retention sequence under *UAS* control. (*UASp-Lys-GFP-KDEL*) | BDSC #31423. Snapp et al (2004) |
| *w[\*]; P{w[+m\*]=betaTub56D-EGFP.I}17-1, H2Av-mRFP1; MKRS/TM6B* | Described in de-Carvalho et al (2022). Originally described in Inoue et al (2004) (Kyoto Stock Center #109603) and (Schuh et al, 2007) (BDSC #23651). |
| *w\*;;Atlastin-EGFP/TM6B EGFP*-tagged knock-in in the C-terminal region of the endogenous locus of atlastin | This study |
| *w[1118]; P{w[+mC]=ncd-Eb1.GFP}M1F3* Expresses *GFP*-tagged *Eb1* protein during oogenesis and early embryogenesis under control of *ncd* regulatory sequences. (*ncd>EB1–GFP*) | BDSC #57327. |
| *w\*;; Rad21*<sup>ex15</sup>*, polyubiq-H2B-RFP, tubpr-Rad21*<sup>(550-3TEV)</sup> *-myc10 (4c)* | Described in Oliveira et al (2010) (internal stock CHR#629) |
| *w\*; ERsp-EYFP-KDEL; Rad21*<sup>ex15</sup>*, polyubiq-H2B-RFP, tubpr-Rad21*<sup>(550-3TEV)</sup> *-myc10 (4c)* | Derived from BDSC #7195 and CHR#629 (see above) |
| *w\*;Rtnl1-GFP; Rad21*<sup>ex15</sup>*, polyubiq-H2B-RFP, tubpr-Rad21*<sup>(550-3TEV)</sup> *-myc10 (4c)* | Derived from BDSC #110579 and CHR#629 (see above) |
| *y[1] w[\*]; P{w[+mC]=UAS-Lam.GFP}3-3* Expresses *GFP*-tagged lamin under UAS control. (*UAS-Lamin–GFP*) | BDSC #7376. |
| *w\*; P{UASp-RFP.KDEL}10/TM3, Sb1 (RFP-KDEL)* | BDSC #30909 |
| *y[1] w[\*]; P{w[+mC]=UAS-Lam.GFP}3-3; P{UASp-RFP.KDEL}10/TM3, Sb1* | Derived from BDSC #7376 and #30909 |
| *w;; G302-Gal4/MKRS* Maternal germline-specific driver | Gift from M Bettencourt-Dias, originally from Daniel St. Johnston; Gurdon Institute, UK. |

## FRAP assays

FRAP experiments were performed using Andor's Mosaic system with a 470-nm laser, using 0.25-s (Lys–GFP–KDEL expressing embryos) or 0.5-s time points (Rtnl1-GFP expressing embryos), and a single z plane was acquired. Collective movement of spindles in the embryo was corrected using the *stackreg* plugin in Fiji (Thévenaz et al, 1998). The first time point was used as a prebleach reference for fluorescence intensity, and bleaching was set to the second time point. Raw fluorescence intensity values were extracted from the time-lapse movies with the plot z-axis profile option in Fiji. FRAP recovery curves were analyzed using the easyFRAPweb tool. Briefly, three different ROIs were measured over time: the bleached region (circle with 42 pixel diameter), the entire ER compartment of a single nucleus (variable size), and the background (region outside the embryo, circle with 95 pixel diameter). ROI positioning was kept constant throughout the time series (this led to occasional inflow of Lys–GFP–KDEL–labelled vesicles, which may slightly overestimate the mobile pool in this strain).

## ER fluorescence intensity

ER mean intensity of control (+buffer) and cytATL-injected embryos was measured for the last time point (t = 10 min) of the time-lapse. A circle ROI with the same area was used to measure the mean intensity at the spindle pole or the equator regions in the ER channel. Relative signal dispersion was calculated by the ratio between the SD and the mean fluorescence intensity (mean).

## Quantification of MT growth and sister chromatid movement

Analysis of EB1 dynamics and chromatid poleward movement was performed based on kymographs created using the FIJI kymograph plug-in (written by J Rietdorf and A Seitz, EMBL, Heidelberg, Germany). We estimated speed by measuring the angle of the linear signals (EB1–GFP for MT growth, histone H2B–mRFP1 for chromatid movement) in the kymographs. Quantification of chromosome movement (Fig S7A) was performed as previously described (Mirkovic et al, 2015). Briefly, H2B–mRFP1 was imaged at 30 s intervals and images were segmented to select the chromosomal regions, based on an automatic threshold (set in the last frame, 10 min after TEV injection). For each binary-image movie, a walking average of three frames was produced (using kymograph plug-in, written by J Rietdorf and A Seitz, EMBL, Heidelberg, Germany), creating a merged image in which the intensity is proportional to the overlap between consecutive frames. Intensity profiles were used to estimate the percentage of non-overlapping, two-frame overlap, and three-frame overlap pixels. The area occupied by sister chromatids (Fig S7B) was calculated using a macro that filters, creates a mask, and subsequently fits a convex hull algorithm enclosing all the sister chromatids. Then, a spline connecting all the

sister chromatids is created. The area of this fitted spline is measured at each time point, estimating the area occupied by sister chromatids at each time frame.

## Distance between sibling and non-sibling nuclei

Measurements of distance between sibling (daughter) and non-sibling nuclei were performed using the line tool in Fiji, 3 min after anaphase onset (counting from the time point of sister chromatid separation). On average, five pairs of daughter nuclei and six non-sibling (neighboring) nuclei were measured. Graphic representation was performed using Prism 7 software (RRID:SCR_002798; GraphPad).

## Statistical analysis

All data sets were first tested for normal distribution using the D'Agostino-Pearson normality test. Data sets that passed the normality test were compared using parametric tests one-way ANOVA (for multiple comparisons) or (un)paired two-sided $t$ test, using Prism (GraphPad). For non-normal data sets, the nonparametric Kruskal–Wallis test was used instead. Details for each comparison can be found on the respective figure legends.

# Data Availability

All the data sets from the present work can be found as supplementary materials.

# Supplementary Information

# Acknowledgements

We thank members of the Telley and Oliveira labs for fruitful discussions. We thank the dedicated facilities at Instituto Gulbenkian de Ciência (IGC): the Fly Facility, the Fly Transgenesis and Genome Editing Facility (FTGEF), the Advanced Imaging Facility (AIF), and the Technical Support Service. We also extend special thanks to Mariana Ferreira (AIF) for help with deconvolution and 3D rendering and Leonardo Guilgur (FTGEF) for the production of Atlastin-EGFP flies. We acknowledge financial support from the European Research Council (ERC-2014-STG 638917-ChromoCellDeV) to RA Oliveira, Fundação para Ciência e a Tecnologia (FCT) supporting M Araújo (PD/BD/128431/2017), RA Oliveira (CEECIND/01092/2017), IA Telley (Investigador FCT IF/00082/2013), and DV Vieira (Project Grant PTDC/BIA-BQM/31843/2017); Fundação Calouste Gulbenkian (FCG) and LISBOA-01-0145-FEDER-007654 supporting IGC's core operation; LISBOA-01-7460145-FEDER-022170 (Congento) supporting the Fly Facility and the Fly Transgenesis and Genome Editing Facility; PPBI-POCI-01-0145-FEDER-022122 supporting the AIF, all co-financed by FCT (Portugal) and the Lisboa Regional Operational Program (Lisboa2020) under the PORTUGAL2020 Partnership Agreement (European Regional Development Fund).

## Author Contributions

M Araújo: conceptualization, formal analysis, investigation, and writing—original draft, review, and editing.
A Tavares and DV Vieira: resources and writing—review and editing.
IA Telley and RA Oliveira: conceptualization, formal analysis, supervision, funding acquisition, and writing—original draft, review, and editing.

## Conflict of Interest Statement

The authors declare that they have no conflict of interest.

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
