## [Reviewer comments · Life Science Alliance]

Life Science Alliance

Endoplasmic Reticulum membranes are continuously needed to maintain mitotic spindle size and forces

Margarida Araújo, Alexandra Tavares, Diana Vieira, Ivo Telley, and Raquel Oliveira

DOI: <https://doi.org/10.26508/lsa.202201540>

Corresponding author(s): *Raquel Oliveira, Instituto Gulbenkian de Ciência and Ivo Telley, Instituto Gulbenkian de Ciência*

Review Timeline:

Submission Date:	2022-05-31
Editorial Decision:	2022-06-24
Revision Received:	2022-09-30
Editorial Decision:	2022-10-20
Revision Received:	2022-10-27
Accepted:	2022-10-28

Scientific Editor: Novella Guidi

Transaction Report:

June 24, 2022

Re: Life Science Alliance manuscript #LSA-2022-01540-T

Dr. Raquel A Oliveira
Instituto Gulbenkian de Ciência
Chromosome Dynamics Lab
Rua da Quinta Grande, 6
Oeiras 2780-156
Portugal

Dear Dr. Oliveira,

Thank you for submitting your manuscript entitled "Endoplasmic Reticulum membranes are continuously needed to maintain mitotic spindle size and forces" to Life Science Alliance. The manuscript was assessed by expert reviewers, whose comments are appended to this letter. We invite you to submit a revised manuscript addressing the Reviewer comments.

Thank you for this interesting contribution to Life Science Alliance. We are looking forward to receiving your revised manuscript.

Sincerely,

B. MANUSCRIPT ORGANIZATION AND FORMATTING:

Reviewer #1 (Comments to the Authors (Required)):

In this manuscript, Araújo and colleagues investigate the impact of ER dysfunction on the mitotic spindle and nuclear/chromatin structure during cell division. To do so they impact on the ER structure through microinjection of the cytosolic domain of the ER membrane fusing protein Atlatin that acts as a dominant negative and inhibits ER membrane fusion, and evaluate how the mitotic spindle structure is altered in live drosophila cells. The authors show that ER structural alteration impacts on the integrity of the mitotic spindle and demonstrate that an intact ER is required through metaphase.

Although the observations reported in this manuscript are very interesting, some points need to be clarified and some issues addressed to further confirm the observed phenotype (5 points)

- The study relies on the use of a single molecular tool provided by J Hu, the Dm cytATL recombinant protein. The authors should use a control protein that bears a mutation preventing ATL dimerization: cytATL(R232Q) to ensure that the observed effects are clearly due to the dominant negative role of cytATL.
- To further document the role of the ER structure in maintenance on mitotic spindle, the authors could use pharmacologic agents that alter ER homeostasis (eg ER stress inducers with rapid action dynamics such as thapsigargin or brefeldin A) and evaluate how they compare with the phenotype observed with cytATL.
- A number of ER proteins have been shown to bind microtubules, including Climp63, p180, and certain members of the REEP family. In order to test whether the interactions between the ER and microtubules impact on the structure of the mitotic spindle the authors could also microinject recombinant proteins/peptides such as the short cytoplasmic amino-terminal region of Climp63 (about 44 amino-acids) or the carboxy-terminal region of REEP1.
- The interaction of ER with F-actin has also been reported as well as the interaction of F-actin with the mitotic spindle, the authors could test the impact of formin inhibitors on the relationship between ER and the mitotic spindle. This could also further document the role of ER interactions with cytoskeleton during cell division. Several F-actin regulators (eg FilaminA) were found to directly bind to 2 sensors of the Unfolded Protein Response (UPR) itself found playing a role in cytokinesis (see also point 2). As such it might be relevant to evaluate how the interaction of PERK and IRE1 with filamin A might impact on mitotic spindle formation.
- At last, the authors nicely show the role of ER structure on the mitotic spindle in drosophila tissue in which cells are tightly connected to each other. How would this be modulated in conditions where cell-cell junctions are not as tight (eg isolated cells). Basically, what is the impact of mechanical forces tensing the cells on the ER/mitotic spindle interaction.

Reviewer #2 (Comments to the Authors (Required)):

Araújo and colleagues have studied the role of the ER in mitotic spindle function. They show that in the Drosophila embryo, the spindles are delimited by ER and that interfering with this organisation affects spindle structure. They present data to indicate that the ER has a physical role in mitosis, rather than, or in addition to, its role in concentrating spindle components. Mitosis researchers have fixated on the spindle and not really considered other factors such as membrane compartments. There are several recent papers showing how the ER is important for chromosome segregation. This current work nicely complements those papers and will be of interest to cell biologists. The added twist is that in a syncytial system like the Drosophila embryo, this organisation of an open-but-not-really-open mitosis, is likely an important way for compartmentalisation (physical or chemical) to occur. I thought this paper was a really nice piece of work. I have some comments for the authors to consider.

Fig 1A (for example) shows that the ER is like an envelope and that the centrosomes are apparently separated off from the spindle. This situation is a bit different from other animal cells where the exclusion zone is a) not as well defined and b) includes the spindle poles. My question is how does this look in 3D? Are the centrosomes really detached from the spindle with ER in-between or is there are gap in this ER "envelope" where the microtubules poke through? Perhaps the authors have some high

mag 3D images to answer this point or maybe the answer (at EM level) is already published. I think some description of the morphological organisation of the spindle + ER, and comparing with other systems (i.e. human cells!) would improve the paper.

The FRAP experiments in Fig 2 are interpreted to report changes in ER morphology. However, FRAP is measuring the mobility of the ER marker within the ER (Ellenberg et al., JCB 1997), as well as changes in morphology. Separating out those two processes is hard. The conclusion is "the ER is continuously undergoing significant reorganization", meaning ER movement, but it is possible that the ER is not moving significantly on this time scale, and any recovery is due to diffusion. Strictly speaking, the experiments in 2B for example show that proteins in the ER at the poles can exchange with the rest of the envelope, but we don't know the exact mechanism. I think the authors should include the diffusion possibility or provide an explanation why they favour ER movement/reorganisation on a <1 minute time scale.

I was puzzled by the measurement in Fig 4G. Half the spindle is photobleached, yet the recovery is 100%. Is it normal to get such full recovery, i.e. is there a half spindle's worth of free tubulin nearby that can exchange? Could the authors comment on this?

In Figure 4C it looks like there is some fragmentation of the centrosomes. At 10:00 there are two dark holes in the ER compared to the earlier time point or to the control. Is this a consistent effect?

Minor points:

- Authors should use the term "co-efficient of variation" to describe the relative signal dispersion metric.

Note added during cross-commenting:

The other two referees raise an important point about controlling for cytATL and cell cycle arrest. In my opinion, points 3 and 4 (and possibly point 2) raised by Reviewer 1 are interesting ways to expand this work in the future but are not necessary to shore up what is reported in the current manuscript. On Point 5 from Reviewer 1, the system is *Drosophila* embryo which is a syncytium, so their comment about inter-cell forces does not apply.

If I can also comment on my own points(!) A bioRxiv preprint from the Cohen-Fix lab was posted after I submitted my review. It describes the "centriculum" in *C. elegans*. The authors could discuss this arrangement and how it relates to their system in answer to the first point I raised.

Reviewer #3 (Comments to the Authors (Required)):

1. Summary

In this research, Margarida Araújo and coworkers investigate the role of the endoplasmic reticulum (ER) in the regulation of mitotic spindle function in *Drosophila melanogaster* syncytial embryos. By using different ER marker proteins, they first show (and extend previous observations) that the ER suffers considerable remodeling throughout mitosis, in correlation with spindle positioning and activity. To determine the participation of the ER in spindle activity, they then develop a system to perturb ER architecture by microinjecting the cytosolic domain of the ER fusogen *Atlastin*, which blocks ER membrane fusion. After showing that this treatment perturbs ER topology -most notably of the domains associated with the spindle poles-, the authors demonstrate that this ER disruption leads to smaller mitotic spindle size, reduced pulling force on chromatids, and inefficient daughter nucleus separation. The authors conclude that the ER membranes that are associated with the mitotic spindle poles are crucial for the maintenance of spindle shape and function. This research provides new relevant information concerning the regulation of ER dynamics during mitosis, and on the contribution of the ER to the activity of the mitotic spindle. Notably, this research shows that distinct ER domains undergo differential remodeling along mitosis, and provide evidence that the domains associated with the spindle poles have a role in spindle functioning.

2. Main points

The experiments performed in this research were carefully conducted and are nicely and clearly presented. The data are of high technical quality and were properly analyzed. Nonetheless, the major conclusions of the research stem from a number of experiments, which are based on the analysis of artificially arrested embryos. Most findings of this research showing that the activity of the mitotic spindle is affected when there are severe defects in ER organization were obtained from embryos, which were artificially arrested in metaphase. While this system provides a very powerful approach to address specific aspects of the cell division cycle and to provide valuable insights into the mechanisms driving this process; it, nonetheless, constitutes an artificial system, which could provide limited information as sole source of evidence. The authors should consider strengthen the conclusions of their research by performing alternative experimental approaches. For example, it would be very important to determine whether the defects in spindle size (and the accompanying alterations) observed in metaphase arrested cells are also produced in nuclei that are normally progressing through mitosis. The authors could evaluate the impact on spindle function of microinjecting the cytosolic domain of *Atlastin* (*cytATL*) in embryos progressing normally through mitosis. Alternatively (or

additionally), an experiment to first disturb the ER to then analyze spindle structure of upon metaphase arrest (i.e., to first microinject cytATL and then induce the metaphase arrest) could be performed. Such experiments should provide important information about the actual impact that the disruption of the ER has on mitotic spindle activity.

3. Additional issues

There are additional points in the manuscript that need clarification:

- 1) The manuscript indicates that in the *Drosophila* syncytial embryos the nuclear envelope breaks down during mitosis; however, it is not completely clear how this process takes place. Since the pattern of nuclear envelope breakdown in syncytial embryos differs from that of a "traditional" open mitosis (as illustrated by the partially disassembly of Lamin B envelope in supplemental figure 1), I suggest to better explain how this process occurs (and to provide pertinent corresponding references). This is important to better interpret the results of the research, especially for the readers who are not fully familiar with *Drosophila*.
- 2) Line 82. The sentence and referencing of line "KDEL is a small peptide sequence that targets proteins to the ER lumen (Frescas et al. 2006)" is not fully accurate. The system employed by Frescas et al (2006) to visualize the ER consists of a GFP containing both, an N-terminal ER signal sequence and a C-terminal KDEL retrieval sequence (Lys-GFP-KDEL, which is actually later utilized in the research). A proper reference sustaining that the KDEL sequence on its own can serve as ER signal sequence should be provided.
- 3) Line 86 (and Fig. S1). "(Lamin-GFP, see Fig. S1, t=00:00)": There is no time 00:00 in Fig S1, please rectify.
- 4) Lines 96-98 (and Fig. 1). "This reduction is not accompanied by a decrease in the perimeter of the ER envelope (Fig. 1D), due to an indentation of the ER envelope at spindle poles (Fig. 1C, inset)": Fig. 1D does not show an analysis of the ER envelope perimeter and there is no inset in Fig. 1C, please rectify.
- 5) Lines 103-106 (and Fig. 1). The text states that "We also observed that both ER reporter proteins localized at the spindle midbody as previously reported (Bobinnec et al. 2003), suggesting that a considerable membrane reorganization occurs at this site (Fig. 1A-B, Telophase, arrow)". However, the legend of Fig. 1 indicates that "Arrowheads show events of ER abscission at telophase" (lines 342-3). Do the arrowheads correspond to the arrows indicated in the text (note that there are no arrows in the figure). Were two different processes intended to be illustrated in the figure?
- 6) Lines 156-159. A bibliographic reference for Atlastin should be provided (Line 156). Is "Kutay et al, 2021" the correct reference for the cytATL experiments in *Xenopus*?
- 7) Line 237-8. "In control embryos... (Fig. 5C, arrow)". Was "Fig. 5B" meant in this sentence?
- 8) Figure 4. Line 401. "Arrowheads highlight spindle pole detachment observed upon..."
There are no arrowheads in the figure, please rectify.

Reviewer #1 (Comments to the Authors (Required)):

In this manuscript, Araújo and colleagues investigate the impact of ER dysfunction on the mitotic spindle and nuclear/chromatin structure during cell division. To do so they impact on the ER structure through microinjection of the cytosolic domain of the ER membrane fusing protein Atlastin that acts as a dominant negative and inhibits ER membrane fusion, and evaluate how the mitotic spindle structure is altered in live *Drosophila* cells. The authors show that ER structural alteration impacts on the integrity of the mitotic spindle and demonstrate that an intact ER is required through metaphase.

Although the observations reported in this manuscript are very interesting, some points need to be clarified and some issues addressed to further confirm the observed phenotype (5 points):

- The study relies on the use of a single molecular tool provided by J Hu, the Dm cytATL recombinant protein. The authors should use a control protein that bears a mutation preventing ATL dimerization: cytATL(R232Q) to ensure that the observed effects are clearly due to the dominant negative role of cytATL.

We thank the reviewer for the excellent suggestion of a valid and important control. We have now added additional experiments with purified cytATL(R192Q) (the equivalent mutation in the *Drosophila* protein, as described in (Bian et al., 2011; Byrnes and Sondermann, 2011; Ulengin et al., 2015). These new experiments are now found in the new Figure S4.

This analysis revealed that upon cytATL^{R192Q} injection, no detectable changes could be seen on both ER envelope size and morphology as well as spindle architecture. We therefore conclude that the dominant effect of cytATL is dependent of its dimerization domain which further supports our conclusion that this fragment deforms ER membranes through dimerization with endogenous protein.

- To further document the role of the ER structure in maintenance on mitotic spindle, the authors could use pharmacologic agents that alter ER homeostasis (eg ER stress inducers with rapid action dynamics such as thapsigargin or brefeldin A) and evaluate how they compare with the phenotype observed with cytATL.

We agree that a comparative analysis with other perturbations on the ER would be valuable and have attempted to perturb the ER using the suggested drugs. However, microinjection of high doses of both thapsigargin or brefeldin A did not cause any detectable defect in ER morphology (see figure below). We found that brefeldin A has no effect on the divisions in these embryonic division. In contrast, thapsigargin causes obvious defects in cell division, however these are not accompanied by detectable changes in ER morphology. It should be noted that thapsigargin is likely causing drastic changes in calcium concentrations, which may have an effect on various mitotic processes that go beyond the ER integrity. Accordingly, we observed that thapsigargin in metaphase arrested embryos causes a stabilization of astral microtubules.

Figure R1: Effect of microinjection of brefeldin A (A) and Thapsigargin (B,C) in cycling (A,B) and metaphase-arrested (*UbcH10^{C114S}* injection) embryos (C).

Given the highly pleiotropic or minimal findings with these two experiments, it remains inconclusive whether or not perturbation on ER homeostasis can alone impose major changes in ER structure in this particular system. Therefore, and in the interest of clarity in the story, we opted for not including them in the revised version.

- A number of ER proteins have been shown to bind microtubules, including Climp63, p180, and certain members of the REEP family. In order to test whether the interactions between the ER and microtubules impact on the structure of the mitotic spindle the authors could also microinject recombinant proteins/peptides such as the short cytoplasmic amino-terminal region of Climp63 (about 44 amino-acids) or the carboxy-terminal region of REEP1.

We agree with the reviewer that these are all very interesting suggestions. Nevertheless, as also pointed out by reviewer #2 in cross-referring comments, we consider these interesting future studies on the subject. Such comparative analysis would require a considerable amount of time for construct generation, protein purification optimization, and functional tests, which is unfeasible within the time-frame of revision expected for this manuscript.

- The interaction of ER with F-actin has also been reported as well as the interaction of F-actin with the mitotic spindle, the authors could test the impact of formin inhibitors on the relationship between ER and the mitotic spindle. This could also further document the role of ER interactions with cytoskeleton during cell division. Several F-actin regulators (eg FilaminA) were found to directly bind to 2 sensors of the Unfolded Protein Response (UPR) itself found playing a role in cytokinesis (see also point 2). As such it might be relevant to evaluate how the interaction of PERK and IRE1 with filamin A might impact on mitotic spindle formation.

We thank the reviewer for the interesting suggestions on potential future studies on the topic. We agree that our findings open further questions on how ER may interact with other cytoskeleton elements during mitosis. However, we trust these suggestions are well beyond the scope of the current manuscript.

- At last, the authors nicely show the role of ER structure on the mitotic spindle in *Drosophila* tissue in which cells are tightly connected to each other. How would this be modulated in conditions where cell-cell junctions are not as tight (eg isolated cells). Basically, what is the impact of mechanical forces tensing the cells on the ER/mitotic spindle interaction.

The reviewer may have missed the fact that *Drosophila* initial embryonic development occurs in a syncytium where nuclei are not even individualized as independent cells. Hence, no cell-cell junctions are present in our experimental model system. We apologize if this particularity was not clear. We have made the following changes to make this point clearer to a non-*Drosophila* specialist:

1. We have referred to the model system of the study at the end of the introduction, highlighting its syncytial nature. You can read the following on page 4 (line 74):

“Here, we used microinjection approaches in syncytial Drosophila early embryos to disrupt acutely ER membranes in a metaphase-arrested state.”

2. We have expanded the discussion section where we further highlight how our findings may be particularly important in context of division without proximal plasma membrane. You can now read the following on page 11 (line 331):

“Anchoring of the centrosome and microtubule aster to the nuclear envelope is particularly relevant during syncytial divisions, as ordered nuclear positioning depends on microtubule asters (Telley et al., 2012; de-Carvalho et al., 2022) (...). Future work should address the role of the ER in other cells where nuclear positioning is critical, including those with canonical anchoring patterns, with direct implications on further organism development.”

Reviewer #2 (Comments to the Authors (Required)):

Araújo and colleagues have studied the role of the ER in mitotic spindle function. They show that in the Drosophila embryo, the spindles are delimited by ER and that interfering with this organisation affects spindle structure. They present data to indicate that the ER has a physical role in mitosis, rather than, or in addition to, its role in concentrating spindle components. Mitosis researchers have fixated on the spindle and not really considered other factors such as membrane compartments. There are several recent papers showing how the ER is important for chromosome segregation. This current work nicely complements those papers and will be of interest to cell biologists. The added twist is that in a syncytial system like the Drosophila embryo, this organisation of an open-but-not-really-open mitosis, is likely an important way for compartmentalisation (physical or chemical) to occur. I thought this paper was a really nice piece of work. I have some comments for the authors to consider.

We thank the reviewer for the nice comments on our work.

Fig 1A (for example) shows that the ER is like an envelope and that the centrosomes are apparently separated off from the spindle. This situation is a bit different from other animal cells where the exclusion zone is a) not as well defined and b) includes the spindle poles.

My question is how does this look in 3D? Are the centrosomes really detached from the spindle with ER in-between or is there are gap in this ER "envelope" where the microtubules poke through? Perhaps the authors have some high mag 3D images to answer this point or maybe the answer (at EM level) is already published. I think some description of the morphological organisation of the spindle + ER, and comparing with other systems (i.e. human cells!) would improve the paper.

We thank the reviewer for this interesting suggestion. We have now re-imaged some embryos at higher spatial resolution in z and obtained 3-D reconstructions. This analysis revealed, as the reviewer rightly pointed out, that the ER forms a circular envelope surrounding each centrosome. This is similar to what was recently

described in *C. elegans* (Maheshwari et al. 2022 bioRxiv). The new results are now presented in figure S2 and discussed on page 5 (line xxx), as follows:

“By metaphase, we found that ER forms a large envelope around the spindle connected to two additional membranous structure surrounding each individual pole (Figure S2), similarly to what has been recently reported in the first division of C. elegans embryos (Maheshwari et al. 2022).”

The FRAP experiments in Fig 2 are interpreted to report changes in ER morphology. However, FRAP is measuring the mobility of the ER marker within the ER (Ellenberg et al., JCB 1997), as well as changes in morphology. Separating out those two processes is hard. The conclusion is "the ER is continuously undergoing significant reorganization", meaning ER movement, but it is possible that the ER is not moving significantly on this time scale, and any recovery is due to diffusion. Strictly speaking, the experiments in 2B for example show that proteins in the ER at the poles can exchange with the rest of the envelope, but we don't know the exact mechanism. I think the authors should include the diffusion possibility or provide an explanation why they favour ER movement/reorganisation on a <1 minute time scale.

We agree with the reviewer that it is hard to conclude whether high mobility reflects high mobility within the ER or re-organization of the membranes. We have favored the second mostly due to prior studies that show a low diffusion of Rtn1. Nevertheless, we agree our experiments are insufficient to draw a solid conclusion. We have therefore rephrased the results to provide both possible interpretations and explain why we favor movement/reorganization.

“The observed high mobile fraction is in contrast to previous studies in yeast and mammalian cells, which detected a much larger immobile fraction for Rtn1/Rtn4a (Shibata et al. 2008), implying that Rtn1 is not an intrinsically diffusible protein within the ER. All together, these findings suggest that Rtn1 is more diffusible in Drosophila early embryos or, more likely, it reflects a high level of reshaping of ER membranes in these embryos.”

I was puzzled by the measurement in Fig 4G. Half the spindle is photobleached, yet the recovery is 100%. Is it normal to get such full recovery, i.e. is there a half spindle's worth of free tubulin nearby that can exchange? Could the authors comment on this?

Syncytial divisions are known for having a large pool of maternally deposited components, which coupled with the syncytial state makes the bleached pool negligible in FRAP studies. In our experience, this is observed on FRAP studies for several dynamic components and not only for the spindle (e.g. chromatin-binding proteins). We have now added a sentence to make this point clearer (page 8):

Such fast recovery rates upon bleaching of a large spindle fraction (half) are consistent with previous studies (REFs), and reflect the high abundance of free tubulin (maternal load) coupled with a fast spindle turnover.

In Figure 4C it looks like there is some fragmentation of the centrosomes. At 10:00 there are two dark holes in the ER compared to the earlier time point or to the control. Is this a consistent effect?

We thank the reviewer for pointing this out. We have quantified the times that two distinct “fragmented” poles are observed in our experiments. We now show that these split poles correspond to disengaged centrioles (Figure S6). Although their separation is not observed across all embryos analyzed, quantitative analysis of the distance between the split poles reveal a higher frequency and increased distances upon *cycATL*-mediated ER disruption.

We concluded that ER membranes surrounding the spindle poles also contribute to centriole engagement.

Minor points:

- Authors should use the term "co-efficient of variation" to describe the relative signal dispersion metric.

We have modified this accordingly.

Note added during cross-commenting:

The other two referees raise an important point about controlling for *cytATL* and cell cycle arrest. In my opinion, points 3 and 4 (and possibly point 2) raised by Reviewer 1 are interesting ways to expand this work in the future but are not necessary to shore up what is reported in the current manuscript. On Point 5 from Reviewer 1, the system is *Drosophila* embryo which is a syncytium, so their comment about inter-cell forces does not apply.

If I can also comment on my own points(!) A bioRxiv preprint from the Cohen-Fix lab was posted after I submitted my review. It describes the "centriculum" in *C. elegans*. The authors could discuss this arrangement and how it relates to their system in answer to the first point I raised.

We now refer to this study twice in the revised manuscript:

- Line (108) we describe the 3D envelope around centrosomes
- Line (327) we discuss our observed changes in centrosome architecture and its potential role as non-canonical anchoring system.

Reviewer #3 (Comments to the Authors (Required)):

1. Summary

In this research, Margarida Araújo and coworkers investigate the role of the endoplasmic reticulum (ER) in the regulation of mitotic spindle function in *Drosophila melanogaster* syncytial embryos. By using different ER marker proteins, they first show (and extend previous observations) that the ER suffers considerable remodeling throughout mitosis, in correlation with spindle positioning and activity. To determine the participation of the ER in spindle activity, they then develop a system to perturb ER architecture by microinjecting the cytosolic domain of the ER fusogen *Atlastin*, which blocks ER membrane fusion. After showing that this treatment perturbs ER topology -most notably of the domains associated with the spindle poles-, the authors demonstrate that this ER disruption leads to smaller mitotic spindle size, reduced pulling force on chromatids, and inefficient daughter nucleus separation. The authors conclude that the ER membranes that are associated with the mitotic spindle poles are crucial for the maintenance of spindle shape and function. This research provides new relevant information concerning the regulation of ER dynamics during mitosis, and on the contribution of the ER to the activity of the mitotic spindle. Notably, this research shows that distinct ER domains undergo differential remodeling along mitosis, and provide evidence that the domains associated with the spindle poles have a role in spindle functioning.

2. Main points

The experiments performed in this research were carefully conducted and are nicely and clearly presented. The data are of high technical quality and were properly analyzed. Nonetheless, the major conclusions of the research stem from a number of experiments, which are based on the analysis of artificially arrested embryos. Most findings of this research showing that the activity of the mitotic spindle is affected when there are severe defects in ER organization were obtained from embryos, which were artificially arrested in metaphase. While this system provides a very powerful approach to address specific aspects of the cell division cycle and to provide valuable insights into the mechanisms driving this process; it, nonetheless, constitutes an artificial system, which could provide limited information as sole source of evidence. The authors should consider strengthen the conclusions of their research by performing alternative experimental approaches. For example, it would be very important to determine whether the defects in spindle size (and the accompanying alterations) observed in metaphase arrested cells are also produced in nuclei that are normally progressing through mitosis. The authors could evaluate the impact on spindle function of microinjecting the cytosolic domain of *Atlastin* (*cytATL*) in embryos progressing normally through mitosis. Alternatively (or additionally), an experiment to first disturb the ER to then analyze spindle structure of upon metaphase arrest (i.e., to first microinject *cytATL* and then induce the metaphase arrest) could be performed. Such experiments should provide important information about the actual impact that the disruption of the ER has on mitotic spindle activity.

We have attempted to address this valid point by various means. However, as outlined below, experimental limitations precluded a clear analysis of ER disruption in cycling (non-arrested) embryos.

To fully bypass the caveat of artificial arrests, we have injected cytATL during interphase and monitor its effect on both ER morphology and nuclear division. However, we observed that in these experimental conditions, the effect of cytATL on ER morphology is very mild, precluding full disruption in the short interphase period between consecutive mitosis.

We trust that this minor effect is likely attributed to major cell cycle differences:

- 1) the ER mobile fraction of ER proteins is higher during mitosis which likely renders membranes easier to disrupt (as shown by our FRAP studies)
- 2) We found that wt-Atlastin is specifically accumulated at mitotic ER (predominantly at centrosomes), when compared with its interphase distribution.
- 3) Atlastin has been proposed to be regulated by cdk1 activity in mitosis (Bergman et al 2015) and the putative Cdk consensus site is located within the cytoplasmic fragment used. It is conceivable that phosphorylation may enhance the disruptive effect of cytATL.

Hence, the slow kinetics of ER disruption in interphase makes our approach unapplicable to the fast nuclear divisions (interphase ~6-8 minutes; mitosis 3-4 minutes) without the artificial arrest in metaphase. To make these cell cycle differences clearer, we have now included new results from a newly established line where endogenous Atlastin was tagged with EGFP (crispR-cas9 mediated genome editing) (Figure S). These results highlight the strong accumulation of endogenous Atlastin at centrosome-proximal regions, particularly in mitosis. We trust these experiments will clarify why the effect is more pronounced in mitosis and at centrosome-proximal regions.

We have also performed the experimental set-up suggested by the reviewer, where cytATL was injected before UbcH10. This resulted in metaphase-arrested spindles that are very similar to the ones reported in our initial submission (cytATL injected specifically at metaphase). However, due to the fast cycles and the slow kinetics of ER disruption in interphase, the observed effect in both ER morphology and spindle size is only evident during the mitosis arrest. This approach is therefore unsuitable to address the reviewer's point.

3. Additional issues

There are additional points in the manuscript that need clarification:

- 1) The manuscript indicates that in the *Drosophila* syncytial embryos the nuclear envelope breaks down during mitosis; however, it is not completely clear how this process takes place. Since the pattern of nuclear envelope breakdown in syncytial embryos differs from that of a "traditional" open mitosis (as illustrated by the partially disassembly of Lamin B envelope in supplemental figure 1), I suggest to better explain how this process occurs (and to provide pertinent corresponding references). This is important to better interpret the results of the research, especially for the readers who are not fully familiar with *Drosophila*.

We thank the reviewer for pointing this out. We have now introduced the fact that *Drosophila* embryos undergo a semi-open mitosis to make this difference clearer to a non-*Drosophila* audience. On page 4 (lines 92) you can read the following:

Note that in contrast to the canonical “open” mitosis (when the NE disassembles completely) Drosophila embryos are an intermediate case, where the NE is only partially disassembled, predominately at the poles (De Souza and Osmani 2007; Katsani et al. 2007; Strunov et al. 2018) and Figure S1). Despite the residual lamin localization as tubular-like structures, the NE is no longer intact after NEBD, and ER is the main continuous membranous structure surrounding the spindle (Figure S1).

2) Line 82. The sentence and referencing of line "KDEL is a small peptide sequence that targets proteins to the ER lumen (Frescas et al. 2006)" is not fully accurate. The system employed by Frescas et al (2006) to visualize the ER consists of a GFP containing both, an N-terminal ER signal sequence and a C-terminal KDEL retrieval sequence (Lys-GFP-KDEL, which is actually later utilized in the research). A proper reference sustaining that the KDEL sequence on its own can serve as ER signal sequence should be provided.

This point has been corrected and references updated.

3) Line 86 (and Fig. S1). "(Lamin-GFP, see Fig. S1, t=00:00)": There is no time 00:00 in Fig S1, please rectify.

These have been corrected.

4) Lines 96-98 (and Fig. 1). "This reduction is not accompanied by a decrease in the perimeter of the ER envelope (Fig. 1D), due to an indentation of the ER envelope at spindle poles (Fig. 1C, inset)": Fig. 1D does not show an analysis of the ER envelope perimeter and there is no inset in Fig. 1C, please rectify.

These have been corrected.

5) Lines 103-106 (and Fig. 1). The text states that "We also observed that both ER reporter proteins localized at the spindle midbody as previously reported (Bobinnec et al. 2003), suggesting that a considerable membrane reorganization occurs at this site (Fig. 1A-B, Telophase, arrow)". However, the legend of Fig. 1 indicates that "Arrowheads show events of ER abscission at telophase" (lines 342-3). Do the arrowheads correspond to the arrows indicated in the text (note that there are no arrows in the figure). Were two different processes intended to be illustrated in the figure?

Both have been corrected. We hope it is clear now we were referring to the same event.

6) Lines 156-159. A bibliographic reference for Atlastin should be provided (Line 156). Is "Kutay et al, 2021" the correct reference for the cytATL experiments in Xenopus?

References have been added to support the claim of fusion activity of Atlastin. The subsequence sentence has also been modified to emphasize the work in human cells.

7) Line 237-8. "In control embryos... (Fig. 5C, arrow)". Was "Fig. 5B" meant in this sentence?

This has been corrected

8) Figure 4. Line 401. "Arrowheads highlight spindle pole detachment observed upon..." There are no arrowheads in the figure, please rectify.

This has been corrected

October 20, 2022

RE: Life Science Alliance Manuscript #LSA-2022-01540-TR

Prof. Raquel A Oliveira
Instituto Gulbenkian de Ciência
Chromosome Dynamics Lab
Rua da Quinta Grande, 6
Oeiras 2780-156
Portugal

Dear Dr. Oliveira,

Thank you for submitting your revised manuscript entitled "Endoplasmic Reticulum membranes are continuously needed to maintain mitotic spindle size and forces". We would be happy to publish your paper in Life Science Alliance pending final revisions necessary to meet our formatting guidelines.

- please add a Discussion section after the Results. Please note that discussion can be combined with Results (resulting in "Results and Discussion" section) for shorter article up to 5 main Figures
- please provide accurate information concerning the use EYFP-KDEL as ER marker as requested by Reviewer 3
- please add ORCID ID for secondary corresponding author-you should have received instructions on how to do so
- please use the [10 author names, et al.] format in your references (i.e. limit the author names to the first 10)
- please add a callout for Figure 5I and Figure S7C to your main manuscript text

Figure Check:

- please add scale bars in Figure S2

A. FINAL FILES:

B. MANUSCRIPT ORGANIZATION AND FORMATTING:

Sincerely,

Reviewer #1 (Comments to the Authors (Required)):

The authors nicely addressed all the comments raised on the initial version of the manuscript

Reviewer #2 (Comments to the Authors (Required)):

The authors have addressed all my comments in full and I have no other concerns.

Reviewer #3 (Comments to the Authors (Required)):

All my concerns have been addressed. Nonetheless, there is still inaccurate information concerning the EYFP-KDEL protein used as ER marker (lines 86-88). The initial targeting of most ER luminal proteins is typically mediated by an N-terminal hydrophobic targeting sequence, the signal peptide (SP). The C-terminal KDEL retrieval sequence is implicated in later "preventing" ER resident proteins to "scape" to the Golgi (more accurately, in retrieving proteins that have escaped to the Golgi, as described in the Bräuer, et al, 2019 reference now provided). In their research, Araujo and coworkers use a YFP that contains only the KDEL retrieval sequence (EYFP-KDEL) as ER marker. As pointed out before, in opinion of this reviewer a proper reference sustaining that the KDEL sequence on its own can serve as ER targeting signal sequence (i.e., in addition to ER retrieval signal) should be provided. Specifically, in the revised version of the manuscript it is stated that "EYFP fused to the KDEL sequence reports all ER" (lines 87-88). However, again, in the reference provided (LaJeunesse et al., 2004), the authors employ a fluorescent protein containing both, an N-terminal ER signal sequence (in this case the ER targeting sequence of human calreticulin) and a C-terminal KDEL retrieval sequence, rather than the latter alone. The authors should provide accurate information concerning the use EYFP-KDEL as ER marker.

Reply to reviewer #3's comment:

All my concerns have been addressed. Nonetheless, there is still inaccurate information concerning the EYFP-KDEL protein used as ER marker (lines 86-88). The initial targeting of most ER luminal proteins is typically mediated by an N-terminal hydrophobic targeting sequence, the signal peptide (SP). The C-terminal KDEL retrieval sequence is implicated in later "preventing" ER resident proteins to "scape" to the Golgi (more accurately, in retrieving proteins that have escaped to the Golgi, as described in the Bräuer, et al, 2019 reference now provided). In their research, Araujo and coworkers use a YFP that contains only the KDEL retrieval sequence (EYFP-KDEL) as ER marker. As pointed out before, in opinion of this reviewer a proper reference sustaining that the KDEL sequence on its own can serve as ER targeting signal sequence (i.e., in addition to ER retrieval signal) should be provided. Specifically, in the revised version of the manuscript it is stated that "EYFP fused to the KDEL sequence reports all ER" (lines 87-88). However, again, in the reference provided (LaJeunesse et al., 2004), the authors employ a fluorescent protein containing both, an N-terminal ER signal sequence (in this case the ER targeting sequence of human calreticulin) and a C-terminal KDEL retrieval sequence, rather than the latter alone. The authors should provide accurate information concerning the use EYFP-KDEL as ER marker.

We agree with the reviewer that initial description was not providing full details on the construct design and was misleading. The reporter we used is not solely EYFP-KDEL but has additionally the signal peptide, as the reviewer rightly pointed out.

We have now changed the way this reporter is introduced in the results section, with the respective reference, to better describe the design of the reporter we used.

We used flies expressing the chromatin marker H2B-mRFP1, and an ER marker with a EYFP flanked by an ER targeting signaling peptide (human calreticulin target sequence) at N-terminus and a KDEL sequence, a small peptide sequence that retains proteins in the ER lumen (Bräuer et al., 2019) at C-terminus, referred therein simply as ERsp-EYFP-KDEL (LaJeunesse et al., 2004).

We have also replaced in all figure the labelling so that you can read ERsp-EYFP-KDEL instead of EYFP-KDEL.

October 28, 2022

RE: Life Science Alliance Manuscript #LSA-2022-01540-TRR

Prof. Raquel A Oliveira
Instituto Gulbenkian de Ciência
Chromosome Dynamics Lab
Rua da Quinta Grande, 6
Oeiras 2780-156
Portugal

Dear Dr. Oliveira,

Thank you for submitting your Research Article entitled "Endoplasmic Reticulum membranes are continuously needed to maintain mitotic spindle size and forces". It is a pleasure to let you know that your manuscript is now accepted for publication in Life Science Alliance. Congratulations on this interesting work.

DISTRIBUTION OF MATERIALS:

Again, congratulations on a very nice paper. I hope you found the review process to be constructive and are pleased with how the manuscript was handled editorially. We look forward to future exciting submissions from your lab.

Sincerely,
